



# Inverse modelling of carbonyl sulfide: implementation, evaluation and implications for the global budget

Jin Ma[1], Linda M. J. Kooijmans[2], Ara Cho[2], Stephen A. Montzka[3], Norbert Glatthor[4], John R. Worden[5], Le Kuai[5], Elliot L. Atlas[6], and Maarten C. Krol[1,2]

[1]Institute for Marine and Atmospheric Research, Utrecht University, Utrecht, The Netherlands
[2]Meteorology and Air Quality, Wageningen University & Research, Wageningen, The Netherlands
[3]National Oceanic and Atmospheric Administration (NOAA), Boulder, CO, USA
[4]Institute of Meteorology and Climate Research, Karlsruhe Institute of Technology, Karlsruhe, Germany
[5]Jet Propulsion Laboratory, California Institute of Technology, Pasadena, California, USA
[6]Rosenstiel School of Marine and Atmospheric Science, University of Miami, Miami, USA

**Correspondence:** Jin Ma (j.ma@uu.nl)

**Abstract.**

Carbonyl sulfide (COS) has the potential to be used as a climate diagnostic due to its close coupling to the biospheric uptake of $CO_2$ and its role in the formation of stratospheric aerosol. The current understanding of the COS budget, however, lacks COS sources, which have previously been allocated to the tropical ocean. This paper presents a first attempt of global inverse

modelling of COS within the 4-Dimensional variational data-assimilation system of the TM5 chemistry transport model (TM5-4DVAR) and a comparison of the results with independent COS observations. We focus on the global COS budget, including COS production from its precursors carbon disulfide ($CS_2$) and dimethyl sulfide (DMS). To this end, we implemented COS uptake by soil and vegetation from an updated biosphere model (SiB4), and new inventories for anthropogenic and biomass burning emissions. The model framework is capable of closing the COS budget by optimizing for missing emissions using

NOAA observations in the period 2000–2012. The addition of 432 Gg S $a^{-1}$ COS is required to obtain a good fit with NOAA observations. This missing source shows little year-to-year variations, but considerable seasonal variations. We found that the missing sources are likely located in the tropical regions, and an overestimated biospheric sink in the tropics cannot be ruled out. Moreover, high latitudes in the Northern Hemisphere require extra COS uptake or reduced emissions. HIPPO aircraft observations, NOAA airborne profiles from an ongoing monitoring program, and several satellite data sources are used to

evaluate the optimized model results. This evaluation indicates that COS in the free troposphere remains underestimated after optimization. Assimilation of HIPPO observations slightly improves this model bias, which implies that additional observations are urgently required to constrain sources and sinks of COS. We finally find that the biosphere flux dependency on surface COS mole fraction may substantially lower the fluxes of the SiB4 biosphere model over strong uptake regions. In planned further studies we will implement this biosphere dependency, and additionally assimilate satellite data with the aim to better separate

the role of the oceans and the biosphere in the global COS budget.



## 1 Introduction

Carbonyl sulfide (COS or OCS) is a low abundant trace gas in the atmosphere with a lifetime about 2 years and a tropospheric mole fraction of about 484 pmol mol$^{-1}$ (Montzka et al., 2007). COS is regarded as a promising diagnostic tool for constraining photosynthetic gross primary production (GPP) of $CO_2$ through similarities in their stomatal control (Montzka et al., 2007; Campbell et al., 2017; Berry et al., 2013; Whelan et al., 2018; Kooijmans et al., 2017, 2019; Wang et al., 2016). COS also contributes to stratospheric sulfur aerosols, which have a cooling effect on climate and hence mitigate climate warming (Crutzen, 1976; Andreae and Crutzen, 1997; Brühl et al., 2012; Kremser et al., 2016). In the recent decades, COS mole fractions in the troposphere have remained relatively constant, which implies that sources and sinks of COS are balanced. Whelan et al. (2018) reviewed the state of current understanding of the global COS budget and the applications of COS to ecosystem studies of the carbon cycle. The most pressing challenge currently is to close the global COS budget.

Previous studies show that substantial emissions of COS are coming from oceanic, anthropogenic, and biomass burning sources, and the largest sinks are uptake by plants and soils (Watts, 2000; Kettle et al., 2002; Berry et al., 2013). Oceanic emissions are thought to be the largest source of COS, both directly and indirectly, due to emissions of $CS_2$ and DMS (Lennartz et al., 2017, 2019), which can be quickly oxidized to COS in the atmosphere (Sze and Ko, 1980). There are considerable uncertainties related to this indirect COS source, with reported yields to COS being $(83 \pm 8)$% from $CS_2$ (Stickel et al., 1993), and $(0.7 \pm 0.2)$% from DMS under low NOx conditions at 298 K (Barnes et al., 1996). Blake et al. (2004) reported anthropogenic Asian emissions for COS and $CS_2$, which appear to have been underestimated by 30–100% due to underestimated coal burning in China (Du et al., 2016). Zumkehr et al. (2018) recently presented a new global anthropogenic emission inventory for COS. The new anthropogenic emission estimates are, with 406 Gg S a$^{-1}$ in 2012, substantially larger than the previous estimate of 180.5 Gg S a$^{-1}$ by (Berry et al., 2013). Another recent study (Stinecipher et al., 2019) concluded that it is unlikely that biomass burning accounts for the balance between sources and sinks of COS, due to the relatively small contribution of biomass burning to the total emissions ($(60 \pm 37)$ Gg S a$^{-1}$).

Suntharalingam et al. (2008) made a first attempt to simulate the global COS budget using the GEOS-Chem model and global-scale surface measurements from NOAA. In order to fit the observed seasonal cycle of COS mole fraction, they had to double the terrestrial vegetation uptake, reduce the southern extra-tropical ocean source, and assume an additional COS source of 235 Gg S a$^{-1}$. Berry et al. (2013) implemented COS in the global biosphere model SiB3. They inferred that, in order to compensate for updated COS biosphere and soil sinks of 1093 Gg S a$^{-1}$, there must be additional COS sources of 600 Gg S a$^{-1}$, which was allocated to the ocean. Glatthor et al. (2015) and Launois et al. (2015) estimated direct COS emissions from the ocean as 992 and 813 Gg S a$^{-1}$, respectively, and also Kuai et al. (2015) hinted to underestimated COS sources from tropical oceans by optimizing sources using one month of COS satellite observations by TES-Aura. However, Lennartz et al. (2017) used COS measurements in ocean water to show that the direct oceanic emissions were much lower (130 Gg S a$^{-1}$) than top-down studies suggested. It is therefore not resolved whether ocean emissions account for the missing source.

In this paper, we address several important open questions concerning the COS budget using inverse modelling techniques, employing the TM5-4DVAR modelling system. We focus on the closure of the COS budget, the contributions of the potential


COS precursors CS$_2$ and DMS, and evaluation of the results with aircraft and satellite observations. In Section 2 we will describe the observations, the implementation of COS, CS$_2$, and DMS in TM5, and the inverse modelling system TM5-4DVAR. In Section 3, we will analyze the results of various inverse model calculations, which are discussed further in Section 4.

## 2 Method

This study aims to close the global COS budget by so-called flux inversions. This technique employs atmospheric measurements to optimize sources and sinks of trace gases such that mismatches between simulations and observations are minimized. In Section 2.1 the observations used in this study are introduced. Section 2.2 will subsequently describe our modelling system, including new emission data sets that have been coupled to the modelling system. The inverse modelling framework is discussed in Section 2.3.

### 2.1 Measurements

#### 2.1.1 NOAA flask data

The NOAA surface flask network provides long-term measurements of COS mole fraction at 14 locations at weekly–monthly frequencies. Most of the stations are located in the Northern Hemisphere (NH), as shown in Figure 1. Although the number of sampling sites is modest, their locations cover most latitudinal regions, and sample over both land and coastal area. The
observational error for each site is relatively small ($< 7$ pmol mol$^{-1}$), therefore we have taken inter-annual variability of COS from Table 1 in Montzka et al. (2007) to represent a fixed observational error upper-limit at each site. In general, the observational error defined in this way varies between 4–10 pmol mol$^{-1}$ in the NH, and between 2–4 pmol mol$^{-1}$ in the Southern Hemisphere (SH). This error is used in the inverse modelling as will be described in Section 2.4.

#### 2.1.2 HIPPO aircraft and NOAA airborne data

Flask data of the HIAPER Pole-to-Pole Observations (HIPPO) experiments (Wofsy, 2011; Wofsy et al., 2017) is used to validate the results of the inverse modelling. There are five HIPPO campaigns conducted from 2009 to 2011 that sampled the COS mole fraction from the North Pole to the South Pole, and from the lower troposphere up to the stratosphere. Three different instruments were used to make measurements of COS during HIPPO. Instrument 2 was used by NOAA to measure COS, and instrument 1 was calibrated consistently with the NOAA calibration standard. Instrument 3 is an independent instrument, but
the probability distribution function of the mole fractions shows that the three instruments are measuring consistent values, with similar averages (see Figure S1). Thus, HIPPO data provides valuable data to check the consistency of the optimized COS budget. The flight routes of the five campaigns are shown in Figure 1. In some numerical experiments, HIPPO data are additionally assimilated to investigate their impact on the optimized COS budget. To investigate this impact on the vertical





distribution of COS, we compared to 2008–2011 NOAA airborne data that are mainly available over North America (Figure 1).
The number of aircraft sites used is 19, and the upper altitude that was typically reached is 8 km.

### 2.1.3   Satellite data

Our inverse modelling results are compared to three independent satellite data sources: TES-Aura, ACE-FTS, and MIPAS. We have selected the period 2008–2011 for the comparison.

NASA's Tropospheric Emission Spectrometer (TES) is a both nadir and limb viewing instrument that flies on the AURA
satellite, which was launched in 2004 (Beer et al., 2001). TES measures the infrared radiation emitted from the Earth and atmosphere in high spectral resolution for 16 orbits every other day. From these spectra, abundances of tropospheric trace gases are retrieved. The COS product used in this study is described in Kuai et al. (2014). The COS retrievals cover the whole vertical column and have less than 1 degree of freedom (DOF) and show maximum sensitivity in the 300–500 hPa region. We will therefore focus our comparisons on total COS columns. To account for the non-uniform vertical sensitivity, we use the
Averaging Kernel (AK) in the model–satellite comparison. As described in Kuai et al. (2014), the AK included in the TES data files is defined in log-space, and should be applied as:

$$\ln(\boldsymbol{\chi_{con}}) = \ln(\boldsymbol{\chi_p}) + \mathbf{A}[\ln(\boldsymbol{\chi_m}) - \ln(\boldsymbol{\chi_p})], \tag{1}$$

where $\boldsymbol{\chi_{con}}$, $\boldsymbol{\chi_p}$, and $\boldsymbol{\chi_m}$ are respectively the convolved, prior, and modeled profiles, and $\mathbf{A}$ is the AK. In Section 3.4 the modeled profiles are convolved with the TES AK ($\boldsymbol{\chi_{con}}$), vertically integrated, and compared to the TES columns.

The Atmospheric Chemistry Experiment Fourier Transform Spectrometer (ACE-FTS) is a high spectral resolution infrared FTS instrument that performs solar occultation measurements, with the aim to sample stratospheric and upper tropospheric profiles of trace gases (Boone et al., 2013). The instrument flies on SCISAT, a Canadian satellite mission for remote sensing of the Earth's atmosphere that was launched in 2003. Its orbit covers tropical, mid-latitude, and polar regions. COS is one of the atmospheric trace gases measured by the ACE-FTS instrument (Koo et al., 2017). ACE-FTS profiles have been compared
to balloon observations and generally showed good agreement, with underestimations smaller than 20% (Krysztofiak et al., 2015). We use product version 3.6 and only observations with quality flag of zero are used. ACE-FTS measures COS within 0–150 km vertically, but the data quality is only sufficient in the upper troposphere and lower stratosphere (UTLS).

The Michelson Interferometer for Passive Atmospheric Sounding (MIPAS) is a Fourier transform spectrometer for the detection of limb emission spectra in the middle and upper atmosphere (Fischer et al., 2008). MIPAS flew on ESA's Envisat
platform that operated between 2002–2012. MIPAS delivers global atmospheric COS profiles in the upper troposphere and stratosphere (Glatthor et al., 2015, 2017). Similar to TES, the MIPAS product provides AKs and prior profiles to facilitate comparisons to modelled profiles, but not in log-space (Stiller et al., 2012):

$$\boldsymbol{\chi_{con}} = \boldsymbol{\chi_p} + \mathbf{A}[\boldsymbol{\chi_m} - \boldsymbol{\chi_p}]. \tag{2}$$

As for most other gases, the prior profile for MIPAS COS retrievals is a zero profile. Eq .2 thus becomes a simple multiplication
of the AK with the modelled profiles. The MIPAS product has been compared to modelled COS distributions (Glatthor et al.,





2015) and ACE-FTS (Glatthor et al., 2017). The latter comparison showed that MIPAS retrieves higher mole fractions around the tropopause compared to ACE-FTS. The MIPAS product has also been compared to airborne measurements of the HIPPO, ARCTAS and INTEX-B campaigns (Supplement of Glatthor et al. 2015). Further MIPAS has been compared to MkIV and SPIRALE profiles (Glatthor et al., 2017). Comparable to the ACE-FTS product, MIPAS COS profiles are only valid in the UTLS.

The retrievals of TES, MIPAS and ACE-FTS v3.6 are provided on 14, 60, and 150 vertical levels in the atmosphere, respectively. We map our modelled COS profiles to these levels using a mass conserving interpolation scheme.

### 2.1.4 Seasonal decomposition

In Section 3.1 we apply a simple seasonal decomposition method to our calculated exchange fluxes. The seasonal decomposi-
125 tion is performed using Python module StatsModels version 0.10. The time series are decomposed into trend, seasonality and noise:

$$\boldsymbol{y}(t) = \boldsymbol{y}_t(t) + \boldsymbol{y}_s(t) + \boldsymbol{y}_r(t), \tag{3}$$

with $\boldsymbol{y}(t)$ the monthly exchange fluxes, and $\boldsymbol{y}_t$, $\boldsymbol{y}_s$, and $\boldsymbol{y}_r$ the trend, seasonal, and residual components, respectively.

### 2.2 Model description

#### 2.2.1 Anthropogenic emissions

We have implemented the anthropogenic emissions based on a recent global gridded emission inventory of COS (Zumkehr et al., 2018). Since we aim to model COS, $CS_2$ and DMS as separated tracers, we disentangled the reported COS emissions into COS and $CS_2$ contributions. Here, we applied an assumed yield of 0.87 (Zumkehr et al., 2018), which means that 1 mole $CS_2$ yields 0.87 mole COS. As a precursor of COS, $CS_2$ reacts with OH to produce COS, and has an atmospheric lifetime of
135 about 12 days (Khalil and Rasmussen, 1984). The converted emissions averaged over the period 2000–2012 are summarized in Table 1. The total anthropogenic COS emissions are on average 343.3 Gg S a$^{-1}$, split between direct COS emissions of 147.5 Gg S a$^{-1}$ and $CS_2$ emissions of 450.2 Gg S a$^{-1}$. This indicates that $CS_2$ is an important precursor of COS. Figure 2 shows time-series of COS and $CS_2$ anthropogenic emissions. COS emissions are dominated by industrial and residential coal sources, while $CS_2$ emissions are dominated by rayon industry and $TiO_2$ production. Moreover, while COS emissions remained
relatively constant in the 2007–2012 period, $CS_2$ emissions show an increasing trend.

While Zumkehr et al. (2018) assumed a molar yield of $CS_2$ to COS of 87%, other reported yields are $(83 \pm 8)$% (Stickel et al., 1993) and 81% (Chin and Davis, 1993). We decided to use a yield of 83% in our modelling, while we used the reported yield of 87% to produce the numbers listed in Table 1. This implies that we introduce slightly less COS in the atmosphere compared to using the Zumkehr et al. (2018) data as direct COS emissions. Note that we apply all categorical emissions or
145 fluxes with a monthly time resolution.





### 2.2.2 Biomass burning emissions

We estimated biomass burning emissions based on the widely used GFED V.4.1 data set (Randerson et al., 2018) for six of the seven emissions categories listed in Table 2. In converting dry mass burned to COS emissions, we used the updated emission factors reported in Andreae (2019). For biofuel use, we base our estimates on the Community Emission Data System (CEDS)

(Hoesly et al., 2018). We calculated COS emissions by first converting CO emissions to dry mass burned, which were converted to COS emissions in a second step. Emission factors are listed in Table 2. In this process we made a distinction between biofuel with and without dung. Dung burning is mainly employed in South Asia (Fernandes et al., 2007) and we applied the dung emission ratios only in the region 60-100°E and 0–40°N. Our biomass burning emissions in the 2000–2012 period are in the range 118–154 Gg S a$^{-1}$ (Figure 2), similar to the emissions used in Berry et al. (2013) (135 Gg S a$^{-1}$). The spatial and

seasonal distribution of the biomass burning emissions averaged over the period 2000–2012 is presented in supplementary Figure S2.

### 2.2.3 Biosphere flux

Our biosphere fluxes are based on simulations with the Simple Biosphere model, version 4 (SiB4) (Haynes et al., 2019). SiB4 calculates the uptake by vegetation based on (Berry et al., 2013). Currently, soil uptake is scaled to the $CO_2$ soil respiration

term, and the implementation of specific COS soil models (Sun et al., 2015; Ogée et al., 2016) is ongoing. The SiB4 model was constrained by a prescribed COS mole fraction of 500 pmol mol$^{-1}$ outside of canopy. Meteorological data that are used as forcing for the SiB4 model are taken from the Modern Era Retrospective Analysis for Research and Applications (MERRA) and are available from 1980 onwards (Rienecker et al., 2011). A spin-up of the model was performed for the period 1850-1979 to reach an equilibrium of the carbon pools. As no MERRA data were available for the spin-up period, the climatological

average of MERRA data over the period 1980-2018 was used as meteorological input for the spin-up period. A final simulation was performed for 1980–2018 with the actual MERRA driver data. The 2000–2018 average flux to the biosphere (vegetation plus soil) amounts to -1053 Gg S a$^{-1}$, in line with estimates using SiB3 (-951 Gg S a$^{-1}$ (Kuai et al., 2015)) and (Berry et al., 2013). The spatial and seasonal distribution of the biosphere uptake is shown in Figure S3. The uptake shows a large seasonal cycle in the NH and large uptake over tropical forests. The biosphere fluxes were deployed on a monthly timescale.

### 2.2.4 Ocean emissions

Climatological ocean emissions of COS and the COS precursors $CS_2$ and DMS are based on Suntharalingam et al. (2008) and Kettle et al. (2002). Large quantities of COS, DMS, and $CS_2$ are emitted from open oceans. The estimated DMS emissions are about 22 Tg S a$^{-1}$, and we note that even if the COS yield from oxidation of DMS is as small as 0.7 % (Barnes et al., 1996), already 156 Gg S a$^{-1}$ COS is formed. The $CS_2$ direct emission from oceans is roughly 195 Gg S a$^{-1}$, yielding 81 Gg S a$^{-1}$

of COS. When the ocean water is cold enough, oceans can turn into a sink of COS instead of a source (Lennartz et al., 2017). Supplementary Figure S4 shows the spatial distribution of the January and July direct and indirect ocean emissions of COS.



Note that our estimates of all COS oceanic emissions as 277 Gg S a$^{-1}$ are substantially smaller than the estimates of 813 Gg S a$^{-1}$ by Launois et al. (2015).

## 2.3 TM5-4DVAR inverse modelling system

We have implemented three tracers (COS, CS$_2$, and DMS) in the inverse modelling framework TM5-4DVAR (Krol et al., 2005, 2008; Meirink et al., 2008). In brief, the TM5 model is used to convert fluxes, collected in state vector $\boldsymbol{x}$, to observations $\boldsymbol{y}$:

$$\boldsymbol{y} = \boldsymbol{H}(\boldsymbol{x}), \tag{4}$$

where $\boldsymbol{H}$ represents the global chemistry transport model TM5. Since the relation between fluxes and observations is currently linear, $\boldsymbol{y} = \boldsymbol{H}(\boldsymbol{x})$ can be written as $\boldsymbol{y} = \boldsymbol{Hx}$. In a flux inversion a cost function is minimized. The cost function has the form:

$$J(\boldsymbol{x}) = \frac{1}{2}(\boldsymbol{x} - \boldsymbol{x}_b)^T \mathbf{B}^{-1}(\boldsymbol{x} - \boldsymbol{x}_b) + \frac{1}{2}(\boldsymbol{y} - \boldsymbol{Hx})^T \mathbf{R}^{-1}(\boldsymbol{y} - \boldsymbol{Hx}), \tag{5}$$

where $\boldsymbol{x}_b$ represents the prior state of the fluxes, and $\mathbf{B}$ and $\mathbf{R}$ are the error covariance matrices of the fluxes and observations, respectively. $\mathbf{B}$ contains the errors assigned to the fluxes, as well as their correlations in space and time (i.e. $\mathbf{B}$ is a non-diagonal matrix). $\mathbf{R}$ contains the errors assigned to $(\boldsymbol{y} - \boldsymbol{Hx})$. These errors are assumed to be uncorrelated and they include, next to

190 the observational errors, also errors related to the process of mapping coarse-scale fluxes $\boldsymbol{x}$ to localized observations $\boldsymbol{y}$. The adjoint of the TM5 model (Krol et al., 2008; Meirink et al., 2008) is used to calculate the gradient of this cost function with respect to all elements in the state vector:

$$\nabla J(\boldsymbol{x}) = \mathbf{B}^{-1}(\boldsymbol{x} - \boldsymbol{x}_b) + \boldsymbol{H}^T \mathbf{R}^{-1}(\boldsymbol{Hx} - \boldsymbol{y}). \tag{6}$$

In all inversions, $\boldsymbol{y}$ is represented by COS observations from the NOAA flask network data (Montzka et al., 2007). Our flux

space, however, in addition to COS emissions, may represent CS$_2$ and DMS emissions from anthropogenic activity and oceans. To map their influence on simulated COS observations $\boldsymbol{y}$, we need to consider chemical conversions of CS$_2$ and DMS to COS. CS$_2$ and DMS are short-lived trace gases, with atmospheric lifetimes of approximately 12 days (Khalil and Rasmussen, 1984) and 1.2 days (Khan et al., 2016; Boucher et al., 2003; Breider et al., 2010), respectively. For CS$_2$ we implemented OH-initiated conversion to COS, while for DMS we simply apply exponential decay with a lifetime of 1.2 days. COS itself is also destroyed

by OH in the troposphere and by photolysis in the stratosphere. For OH, we use monthly varying climatological OH fields (Spivakovsky et al., 2000), and applied a correction factor of 0.92 (Naus et al., 2019). In summary, the chemistry that is implemented therefore consists of the following four reactions:

$$\text{COS} + \text{OH} \xrightarrow{r_1} \text{products} \tag{R1}$$

$$\text{COS} + \text{h}\nu \xrightarrow{j_1} \text{products} \tag{R2}$$





$$CS_2 + OH \xrightarrow{r_2} f_1\, COS\ +\ \text{other products} \tag{R3}$$

$$DMS \xrightarrow{r_3} f_2\, COS\ +\ \text{other products}, \tag{R4}$$

where $j_1$ is the stratospheric photolysis frequency, and $r_1$ and $r_2$ are the rate constants of COS and $CS_2$ OH-oxidation, respectively. The fractions $f_1$ and $f_2$ represent the molar yields of COS from $CS_2$ (taken as 0.83 (Stickel et al., 1993)) and DMS (taken as 0.007 (Barnes et al., 1996)). The rate $r_1$ is calculated according to the Arrhenius equation:

$$r_1 = A e^{\frac{-1200\,\mathrm{K}}{T}}, \tag{7}$$

where $T$ is temperature in K , and $A$ is $1.13\times 10^{-12}$ cm$^3$ s$^{-1}$ (Cheng and Lee, 1986). The rate $r_2$ is $2.0 \times 10^{-12}$ cm$^3$ s$^{-1}$. Rate $r_3$ represents an exponential decay of 1.2 days for DMS ($r_3 = 9.6 \times 10^{-6}$ s$^{-1}$).

COS photolysis frequencies are calculated based on troposphere ultraviolet and visible (TUV) radiation model (Madronich et al., 2003). Based on monthly climatologies of ozone profiles and temperatures, monthly-averaged photolysis frequencies are calculated on a 1 km grid spanning 0 – 120 km and on 180 latitude bands. Implemented in TM5, COS loss in the stratosphere amounts to about 40 Gg S a$^{-1}$. This estimate is in line with earlier estimates of (50±15) Gg S a$^{-1}$ (Brühl et al., 2012; Barkley et al., 2008; Chin and Davis, 1995; Engel and Schmidt, 1994; Weisenstein, 1997; Krysztofiak et al., 2015; Turco et al., 1980; Crutzen and Schmailzl, 1983; Crutzen, 1976).

## 2.4 Model-data mismatch errors

The diagonal elements of the error covariance matrix $\mathbf{R}$ in Eq. 5 contain contributions from observational errors, representation errors and errors related to applying large fluxes in the planetary boundary layer (Bergamaschi et al., 2010):

$$\sigma_t = \sqrt{(\sigma_o^2 + \sigma_r^2 + \sigma_f^2)}, \tag{8}$$

where $\sigma_t$ is the total error, $\sigma_o$ the observational error, $\sigma_r$ the representation error, and $\sigma_f$ an error related to applying large surface fluxes. The assumed observational error is shown in Figure 3. It is worth to note that observational errors are usually overwhelmed by the representation and flux errors. The representation error is calculated by sampling the modelled gradients in the vicinity of the sampled station (Bergamaschi et al., 2010). Finally, the flux error in each cell is linked to the magnitude of the monthly surface flux $f$ (kg m$^{-2}$ s$^{-1}$ in each cell) applied in the model as:

$$\sigma_f = \frac{f g M_{\mathrm{air}} \Delta t}{M_S \Delta p}. \tag{9}$$

Here, $f$ represent the sum of all COS prior flux components. In this sum, the biosphere flux is dominant over regions with strong biosphere uptake. Further, $g$ is gravitational acceleration (9.8 m s$^{-2}$), $M_{\mathrm{air}}$ is molar mass of dry air (28.9 kg kmol$^{-1}$),



$M_S$ is molar mass of sulfur (32.1 kg kmol$^{-1}$), $\Delta p$ (Pa) is the thickness of the first model layer, and $\Delta t$ is the time (s) over which the COS flux accumulates (we use 1 hour). Note that $\sigma_f$ is unit-less and is multiplied by $1 \times 10^{12}$ to obtain units of pmol mol$^{-1}$.

Based on the total error, we define a $\chi^2$ metric to quantify how well the observations are reproduced by the model (e.g. at a particular station).

$$\chi^2 = \frac{\sum_{i=1}^{N} (\boldsymbol{Hx} - \boldsymbol{y})^2}{N\sigma_t^2}, \tag{10}$$

where $N$ is the number of individual observations. We can calculate this metric before optimization (prior) and after optimization (posterior). $\chi^2$ is used to diagnose whether inversions are over-fitting or under-fitting the information contained in the measurement network. A value $\chi^2 \approx 1.0$ indicates that the inverse system is able to fit the data within the error setting (Hooghiemstra et al., 2011). A large posterior $\chi^2$ indicates that the state does not have enough degrees of freedom to fit the observations properly (or the error settings are too small). A small posterior $\chi^2$ indicates over-fitting of the observations (or too wide error settings).

## 2.5 Model settings

In this study, the TM5-4DVAR system is employed on a global resolution of $6° \times 4°$ (longitude $\times$ latitude). Flux fields are coarsened from a resolution of $1° \times 1°$. To create a reasonable start field for the inversions, we initially performed an 11-year forward simulation starting with zero initial mole fractions and baseline surface fluxes augmented by 432 Gg S a$^{-1}$, distributed uniformly to close the global budget. After 11 years, sources and sinks are roughly in balance, with atmospheric mole fractions of about 500 pmol mol$^{-1}$. Note that fluxes are used as zero-order terms, while the COS removal by OH and photolysis are first order removal terms that grow as the atmospheric COS increases.

We will present the results of four inversions. Firstly, we optimized the missing emissions, which amount to 432 Gg S a$^{-1}$. This inversion will be denoted by $S_u$ throughout the paper. The aim of this inversion is to investigate the spatial structure and temporal variability of the missing COS emissions. This is the first time that a formal 4DVAR approach is applied to the COS budget. To this end, we start from an emission field of 432 Gg S a$^{-1}$ that is uniformly distributed globally. We optimize emissions on a monthly timescale, and assign a grid-scale prior error of 100%, which is an arbitrary number to give fluxes enough freedom to adjust. In a three-year inversion, the total number of state vector elements amounts to 97200 (36 months$\times$45 latitudinal bins $\times$60 longitudinal bins). The total number of NOAA observations is much smaller, thus rendering the inversion under-determined. We therefore also use inversion $S_u$ to explore different settings of the temporal and spatial correlation lengths, which control the degrees of freedom of the state vector. We explore spatial correlation lengths of 1000 km, 4000 km, 6000 km, 10000 km, and 20000 km, and temporal correlation lengths of 5.5, 7, 9.5, 12 months.

Secondly, we explore the optimization of specific categories in inversions S1–S3. In S1 we attempt to perform an "objective" inversion, in which we assign grid-scale errors of 50% to the biosphere and ocean (we optimize both COS and CS$_2$), and 10% to the anthropogenic COS and CS$_2$ emissions, and to the biomass burning emissions. Furthermore, in S2 we only optimise ocean exchange, and in S3 we only optimise the biosphere exchange. The aim of inversions S1–S3 is to explore whether either



ocean fluxes or the biosphere fluxes (or both) should be used to close the COS budget. Note that DMS ocean emissions are not optimized. The names and setting of the inversions are summarized in Table 3.

The cost function is minimized with Congrad, an efficient numerical algorithm for solving linear systems (Lanczos, 1950).
This minimizer was also used in previous inverse modelling studies with the TM5-4DVAR system (Basu et al., 2013; Monteil et al., 2011, 2013; Houweling et al., 2014; Pandey et al., 2015). For convergence, we request a gradient norm reduction of $1 \times 10^5$, and this reduction is usually achieved within 40 iterations.

We perform flux inversions for the period 2000–2012. To decrease computational costs, we adopt the strategy to run parallel 3-year inversions, and we discard the optimized fluxes of the first 6 months (spin-up) and the last 6 months (spin-down). For
example, the first inversion targets the period 1-1-2000 to 1-1-2003, the second inversion 1-1-2002 to 1-1-2005, and so on. In the spin-up period the fluxes in the first 6 months are used to adjust the imperfect initial condition. In the spin-down period, fluxes are less reliable, because they have not been well constrained by observations. The optimized fluxes in the overlapping years are used to check the inversion results for consistency. In general, it is found that the optimized fluxes in the overlapping periods are highly consistent.

## 3   Results

### 3.1   Closing the COS budget

In this section, we consider inversion $S_u$, in which a uniform field emitting 432 Gg S a$^{-1}$ is optimized. We use different settings for the spatial and temporal correlation lengths of this field in the inversion, and quantify the posterior goodness-of-fit using the $\chi^2$ metric (Eq. 10). As presented in more detail in Supplementary Figure S5 we find, as expected, that $\chi^2$ decreases with
increasing degrees of freedom (smaller correlations).

Overall, the posterior fit to NOAA observations does not improve significantly for smaller correlation lengths. If we analyse the posterior fit to independent data from HIPPO, however, we find that the $\chi^2$ reaches a minimum (see supplement Figure S5). After this minimum, $\chi^2$ values increase again, a possible sign of over-fitting. We therefore select 4000 km and 12 months for respectively the spatial and temporal correlation length, and use these values in the remainder of this study.

Figure 4 presents the fit to observations of the prior and posterior simulation, for the inversion with temporal and spatial correlation lengths of 12 months and 4000 km, respectively. Corresponding $\chi^2$ metrics per station are listed as labels in Figure 4. Posterior fits are by design much better than prior fits. Only for NOAA stations THD and NWR the posterior $\chi^2$ remains larger than 3, indicating insufficient degrees of freedom to resolve remaining discrepancies, underestimated model errors, or the influence of outliers (see Figure 4 g, h). THD is a coastal site (107 m a.s.l.) and NWR is a tundra site above
treeline (3526 m a.s.l.) in the US (Figure 1), and thus the model resolution of 6°×4° might be too coarse to represent these sites. Indeed, observations regularly exhibit very low mole fractions during the growing seasons. It is worth to note that the posterior simulation does not exhibit jumps in overlapping years from the parallel running inversions, indicating that our inversion strategy works well.



The correlation settings have a large impact on the optimized fluxes. Figure 5 shows the spatial distribution of the posterior
flux field calculated with two different correlation settings. For correlations of 1000 km and 5.5 months (panel (a)) we detect
a typical pattern that signals over-fitting of the observations. In such a pattern, the optimized flux displays hot spots close
to measurement locations (e.g. THD, MLO, SMO). For very long spatial correlations, e.g., 20000 km, posterior fits are poor
($\chi^2 > 6$, see Figure S5) and optimized flux patterns show irregular behaviour (Figure S6). Our best inversion (4000 km and
12 months) produces a smooth optimized flux without apparent spatial patterns near observational stations (Figure 5b). This
pattern confirms the missing COS sources in the tropics (Suntharalingam et al., 2008; Berry et al., 2013) and also requires more
uptake at high latitudes, especially in the NH.

To investigate the variation in the optimized fluxes of inversion $S_u$, we decompose the flux components as described in
Section 2.1.4. The monthly fluxes and derived long-term trend are shown in Figure 6. Note that only the "unknown" category
(panel e) has been optimized, and that yearly repeating ocean fluxes are not included in the figure. While there is no clear
discernible trend in the prior biosphere and biomass burning fluxes, the prior anthropogenic emissions show a small increasing
trend in 2003–2007 for COS and in 2010–2012 for $CS_2$. The optimized field shows an irregular seasonal cycle with substantial
inter-annual variability, but no clear long-term trend. This indicates that the "missing sources" are likely partially related to
fluxes with strong seasonal cycles. In the next section, we will therefore explore the optimization of the ocean and biosphere
fluxes.

**3.2 Objective inversions**

In this section we will discuss the results of inversions S1, S2, and S3. The resulting global budgets are compared to literature
values in Table 4. In addition, $\chi^2$ metrics and biases of the various inversions are reported in Table 5 for the NOAA surface
network, the HIPPO campaigns, and the NOAA airborne profiles. Note that we also report results for optimizations that
assimilated the HIPPO observations next to the NOAA surface data. The period of the analysed inversions is 2008–2010.
The prior and posterior emission errors and error reduction of the different inversion scenarios are listed and discussed in
Supplement Table S1.

The three inversions are all able to close the global COS budget, however, with very different budget terms (Table 4).
Inversions S1 and S3 close the COS budget by a drastic reduction of the biosphere uptake in the tropics and more biosphere
uptake at high latitudes. When the biosphere is not optimized (S2), the inversion enhances the $CS_2$ tropical oceanic source and
reduces direct COS emissions from the high latitude oceans (Table 4). Both patterns lead to enhanced release from the tropics
and more uptake at high latitudes, as was found for inversion $S_u$.

Concerning the posterior fit to observations, none of the S1–S3 inversions performed like inversion $S_u$. The statistics in
Table 5 show that $S_u$ leads to the best fit to the assimilated observations, and only a small remaining bias. Inversions S1 and
S3 show better $\chi^2$ statistics and smaller biases than inversion S2, because it is difficult to fit continental NOAA stations (LEF,
HFM, NWR, THD) only by optimizing ocean fluxes. However, S1 and S3 show a tendency to turn the tropical biosphere sink
into a source, as shown in Figure 7, which depicts the posterior biosphere flux and flux increment for inversion S1. Note that
while the uptake over high NH latitudes is enhanced, fluxes over regions in South America and over Indonesia have turned





into a source. This behaviour can be explained by the under-determined nature of the inverse problem: there are simply not enough observations to constrain the tropical fluxes. Fast mixing in the tropics further complicates the detection of signals from the tropical biosphere using the NOAA surface network. Without additional observations it is therefore hard to unequivocally close the tropical COS budget. Currently, inversion S1 mostly assigns the missing sources to reduced biosphere uptake in the tropics, but the superior $S_u$ inversion assigns the missing COS sources to a broad band in the tropics, without strong preference for land or ocean. Note that the behaviour of inversions S1, S2, and S3 is strongly driven by the predefined spatio-temporal patterns in the prior flux fields.

Although we currently cannot close the global COS budget with one specific known flux, it is instructive to explore the information content of independent COS observations. In the next section, we will therefore evaluate the results of our inversions with HIPPO and NOAA airborne observations (Figure 1).

### 3.3 Evaluation with HIPPO and NOAA airborne profiles

From Table 5 it is clear that for all inversions the comparison to HIPPO observations is not very favourable. Most notably, the simulations with optimized fluxes show strong negative biases, and poor $\chi^2$ statistics. However, Figure 8 shows that the inversions S1 and $S_u$ (blue lines) largely improve the correspondence to HIPPO campaign 1 observations (red), relative to the prior simulation (black). The posterior simulations do capture the HIPPO observations much better. The remaining differences in the middle panels of Figure 8 show the general underestimation of the model. However, inversion S1 overestimates HIPPO in the southern tropics, likely caused by too large flux adjustments over South America, the region sampled by HIPPO campaign 1.

Interestingly, when the HIPPO observations are additionally assimilated in the inversion, biases are largely removed (Figure 8, lower panels) while the correspondence to the NOAA surface network deteriorates only slightly (Table 5). Posterior $\chi^2$ values for the HIPPO campaigns remain relatively poor, however, signalling too strict error settings or processes that are not properly modeled.

From the comparison with HIPPO we find that our state vector has enough flexibility to fit additional observations, and that the inversions are strongly observation-limited. Moreover, we find that the inversions based on only observations from the NOAA surface network tend to underestimate COS in the free troposphere. This is corroborated by observations from the NOAA airborne profiles, which are mostly collected over the US (see Figure 1). Figure 9 shows a comparison between profiles using results of inversion S1. Although most posterior profiles (blue) improve considerably compared to the prior simulation (black), they still underestimate observations (red) in the free troposphere. Note that the simulations based on inversion S1 correctly predict the draw-down of COS towards the surface for most measured profiles, and especially the match with the LEF site is very good at the surface, which confirms the performance of the inversion. If HIPPO observations are additionally assimilated (green), the agreement in the free troposphere slightly improves. For S1, $\chi^2$ for the profile comparison reduces from 27.7 to 20.1, and the bias reduces from -13.9 to -9.7 pmol mol$^{-1}$ (Table 5). This confirms the low bias of the free troposphere COS mole fractions in simulations with fluxes that are optimized using both NOAA surface and HIPPO observations.





It is now clear that inversions using surface data from the available NOAA network sites will not be able to separate various source categories, and specifically not in the data-void tropics. In the next section we will therefore investigate the prospects of using satellite data to constrain fluxes.

### 3.4 Satellite validation

In Figure 10 we present a comparison between MIPAS, ACE–FTS, and co-sampled TM5 COS profiles. The latitude–height distributions of MIPAS, TM5 (convolved with the MIPAS AK) and ACE–FTS are shown in Figure 10(a-c). In Figure 10(d) we show inversion S1 (blue), inversion S1 with additional assimilation of HIPPO profiles (green), and averaged TM5 profiles convolved with the MIPAS AK of the prior (black) collocated with respect to ACE–FTS profiles.

In general, TM5 reproduces the observed pattern of COS well, but with lower values in the tropical up-welling region around
375 25 km altitude. The comparison between ACE–FTS and MIPAS is consistent with findings of Glatthor et al. (2017), who found that ACE-FTS is systematically lower in the UTLS region. Moreover, they found that MIPAS data showed no bias compared to MkIv and SPIRALE COS balloon profiles, which also exhibit higher COS values than ACE-FTS (Krysztofiak et al., 2015; Velazco et al., 2011). TM5 profiles, after convolution with the MIPAS AK, are in between MIPAS and ACE–FTS. Prior TM5 profiles (black) show highest values around the tropopause. Again, TM5 profiles optimized by HIPPO and NOAA observation
(green dashed line in Figure 10d) show a slight increase in the upper troposphere compared to the optimization with only NOAA surface-site data (blue dashed line).

TM5 results are also compared to the nadir viewing TES instrument. To this end, COS columns of TM5 (convolved with the TES AK, see Equation 1) and TES are averaged in 20 latitudinal bins between 32°S and 32°N. Outside this latitude band, TES observations become too noisy for a reasonable comparison. Comparisons are shown for the months March, June,
September and December in Figure 11, based on inversion S1, and averaged over the years 2008–2011. This comparison shows that the prior simulation is too high in the tropical latitudes and on the NH (e.g., June, September, and December). After assimilation, the agreement with TES improves, but now a general underestimate can be observed. The inversion in which also the HIPPO observations are assimilated bring the simulated mole fractions closer to TES (except for September), confirming our earlier findings based on the airborne observation. Thus, although the TES-derived columns are rather noisy,
they offer good perspective to better constrain the COS budget in the tropics. Due to the sensitivity of TES to COS in the middle troposphere (Kuai et al., 2015), the assimilation of TES in our 4DVAR system might be able to differentiate between the biosphere and ocean signal, something that turned out to be difficult using NOAA surface observations only.

### 3.5 Discussion

In this study we have presented inversions focused on the closure of the global COS budget. In general, our inversion modelling
framework based on the TM5-4DVAR system is quite well capable to close the global budget (e.g., inversion $S_u$, S1 and S3) and to optimize flux fields such that surface observations are well reproduced. However, due to the lack of observations, we are unable to unambiguously assign the missing COS sources to either missing ocean emissions or to reduced tropical uptake by the biosphere. Firstly, the total number of observations remains relatively small, which leads to an under-determined inversion





problem. Secondly, there are no observational sites that sample air masses from tropical Africa, South America, and South East
Asia, which are regions with important COS fluxes. An important next step will therefore be the utilization of satellite data
in future inverse modelling studies. In the current study, we did not include all exchange fluxes that are reported in literature
(Whelan et al., 2018). In general, we find that our inversions still underestimate COS in the free troposphere. Here, there might
be a role for volcanic emissions (25-42 Gg S a$^{-1}$ (Whelan et al., 2018)), or 'unnoticed' tropical sources like wetland exchange
(-150 to 290 Gg S a$^{-1}$ (Whelan et al., 2018)). Volcanic emissions are important to mitigate the stratospheric aerosol loading in
the stratosphere (Sheng et al., 2015) and might be able to reduce the gap between modelled COS by TM5 and measurements.
Alternatively, missing COS could come from an atmospheric oxidation process that converts $CS_2$ or DMS to COS. We did not
find strong evidence for enhanced $CS_2$ emissions from tropical oceans in our S1 inversion, although inversion S2 produced
reasonable COS simulations by optimizing only COS and $CS_2$ emissions from the ocean. Moreover, our "best" $S_u$ inversion
produced a flux field that indicated enhanced tropical sources over both land and ocean (Figure 5). Thus, field studies that
address tropical COS exchange processes are urgently needed (Lennartz et al., 2020).

The use of COS as a proxy for gross primary productivity needs a better level of understanding of the biosphere flux. Here
we used monthly prior flux fields calculated with the SiB4 model (Berry et al., 2013) in which soil exchange and vegetation
uptake are combined. In future studies, we might need a better prior description of this important global COS sink. For instance,
recent studies (Ogée et al., 2016; Sun et al., 2018; Meredith et al., 2019; Spielmann et al., 2020) stress the importance of the
soil-atmosphere COS exchange. Our inversions S1 and S3 calculate large increments in the biosphere exchange (Figure 7),
with general less uptake in the tropics (turning the flux even into a COS source) and enhanced uptake in the NH high latitudes.
Quantitatively, the COS uptake is reduced from a prior value of 1053 Gg S a$^{-1}$ to 557 Gg S a$^{-1}$ to close the COS budget. While
we seriously question the validity of this result given the fact that most flux adjustments are projected in the data-void tropics,
it is still instructive to consider the feedback of the atmospheric COS mole fractions on COS uptake. Since biosphere models
operate mostly uncoupled to atmospheric transport models, we used a fixed mole fraction of 500 pmol mol$^{-1}$ to construct the
prior biosphere fluxes. However, observations clearly show a large drawdown of COS near the surface (Hilton et al., 2017;
Spielmann et al., 2020). We therefore explored the calculations in SiB4 and found that biosphere flux should scale linearly
with atmospheric COS mole fractions (Berry et al., 2013). To estimate the potential impact of reduced mole fractions at the
surface on the biosphere flux, we corrected the monthly SiB4 fluxes as:

$$f_{\mathrm{biosp,cor}} = f_{\mathrm{biosp}} \frac{y(\mathrm{COS})}{500 \, \mathrm{pmol \, mol}^{-1}}, \tag{11}$$

where $f_{\mathrm{biosp}}$ and $f_{\mathrm{biosp,cor}}$ are the original and corrected monthly biosphere fluxes on the TM5 grid, and $y(\mathrm{COS})$ is the
monthly mean COS mole fractions (pmol mol$^{-1}$) in the first model layer (approximately 50 m) from inversion $S_u$. This simple
correction, based on monthly mean fields, changes the biosphere sink from –1053 Gg S a$^{-1}$ to –851 Gg S a$^{-1}$, an update of
202 Gg S a$^{-1}$ (Supplementary Figures S7, S8 and S9). Interestingly, the corrected flux is strongly reduced over regions with
an active tropical biosphere, in line with results from inversions S1 and S3. This indicates that uptake of COS should be treated
as a first order loss process, and that the SiB4 prior fields based on fixed atmospheric mole fractions of 500 pmol mol$^{-1}$ likely





overestimate COS uptake. However, such an approach makes the optimization problem non-linear. This, and the challenge of assimilating satellite observations, will be the subject of future studies.

## 4 Conclusions

In this study, we have implemented an inverse modelling framework for COS, coupled to the budgets of $CS_2$ and DMS. Inversions using the NOAA surface observation network have been evaluated with observations from HIPPO, airborne observations, and satellite products. Conclusions are:

- In line with earlier studies, our inversions point to missing sources in the tropics and missing sinks at high latitudes. We identify that the flux field that closes the budget exhibits an irregular seasonal cycle. Whether the missing sources in the 440 tropics originate from the land or ocean cannot be determined currently, because observations in the tropics are sparse. Part of the tropical sources can be explained by the dependence of COS uptake on atmospheric mole fractions.

- Simulations that are optimized by only NOAA surface observations lack information about COS in the free troposphere. When HIPPO observations are used as an additional data source in the inversions, the comparison to NOAA airborne observations and satellite products generally improves.

- Future improvements are expected from the assimilation of satellite data and better prior descriptions of the ocean and biosphere fluxes.

Our future plan is therefore to assimilate satellite data into our 4DVAR inverse modelling system to have better constraints on COS in the free troposphere and lower stratosphere. Other developments target the coupling of COS and $CO_2$ in a shared inverse modelling system, with the aim to better constrain gross primary productivity.

*Code availability.* Codes are available on request from authors.

*Author contributions.* JM and MK designed the study. JM implemented the method and carried out numerical experiments. LMJK and AC performed SiB4 model experiments. SM and EA provided observations from the NOAA surface network, HIPPO and airborne observations. JW and LK provided the TES–Aura satellite product and helped with the technical discussion on the use of satellite products. NG provided the MIPAS satellite data product with averaging kernel and helped with the discussion. JM and MK wrote the manuscript with contributions 455 from all co-authors.

*Competing interests.* There is no competing interests from authors.





*Acknowledgements.* We thank the NOAA team providing the data and help from Dr. Lei Hu for providing the airborne profiles. We thank the MIPAS research team for providing MIPAS satellite data product. We thank for data support from the HIPPO team, and the TES and ACE-FTS satellite data teams. Elliot Atlas acknowledges support from NSF AGS Grant 0959853. This research project has been supported 460 by funding from the European Research Council (ERC) under the European Union's Horizon 2020 research and innovation program under grant agreement No 742798 (http://cos-ocs.eu).



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

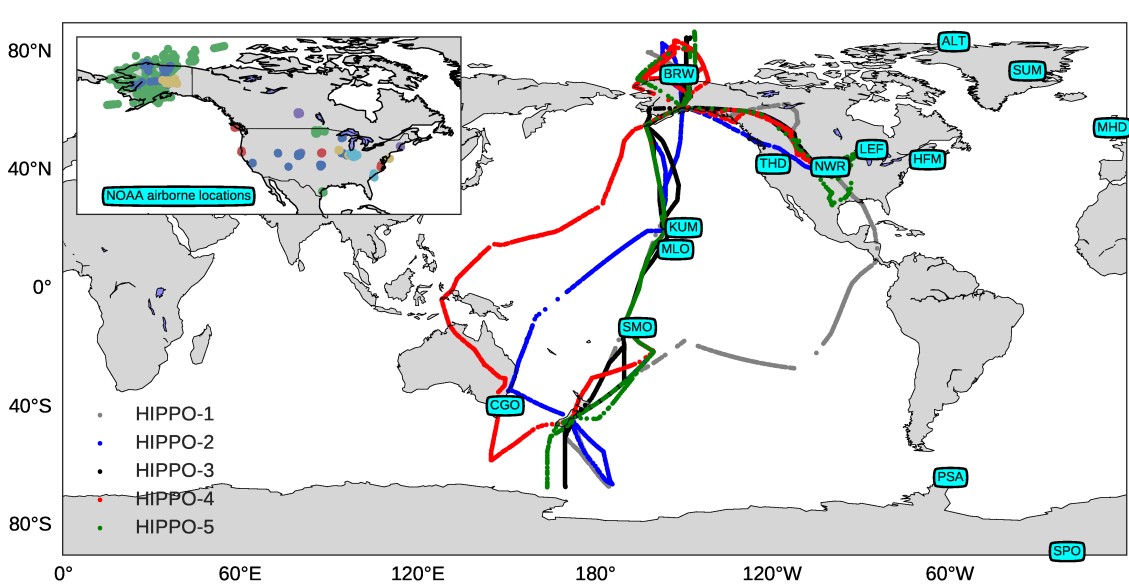

**Figure 1.** Geographical locations of the NOAA ground-based observations (shown in boxes), the five HIPPO campaign tracks, and the NOAA profile program (inset). Note that there are no NOAA surface stations located in Asia, South America, Africa or Europe.

**Figure 2.** Yearly anthropogenic emissions of COS and $CS_2$ and COS biomass burning emissions in the period 2000 to 2012. We disentangled the emissions reported in Zumkehr et al. (2018) into COS and $CS_2$ emissions using their reported yield of 0.87 (see main text). Biomass burning emissions are calculated based on the GFED 4.1 biomass burning inventory and the CEDS biofuel emission inventory (see main text).



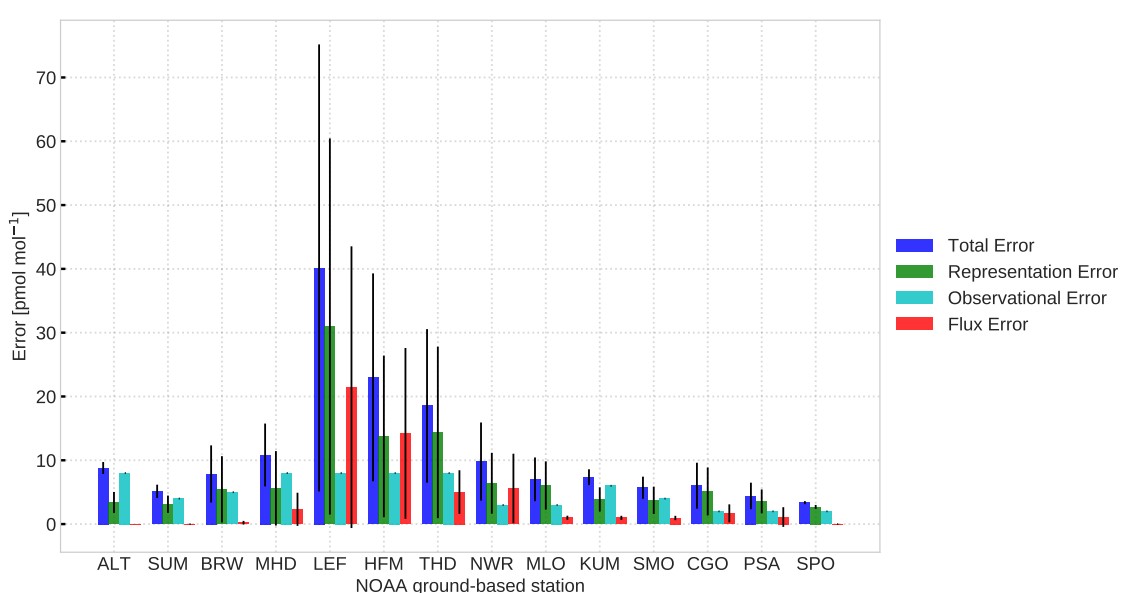

**Figure 3.** Error analysis for NOAA stations. Black error bars represent the time-variations of the errors over a 3–year period ( 2008–2010). For ALT, SPO and SUM, the flux-related errors are close to zero and not shown. Stations are ordered from the North Pole to the South Pole.





**Figure 4.** COS prior and posterior comparison at NOAA stations. Red dots and bars are NOAA measurements with errors. Blue and black dots represent the posterior and prior simulation, respectively. Results are shown for inversion $S_u$ in which only the "unknown" emission category is optimized.

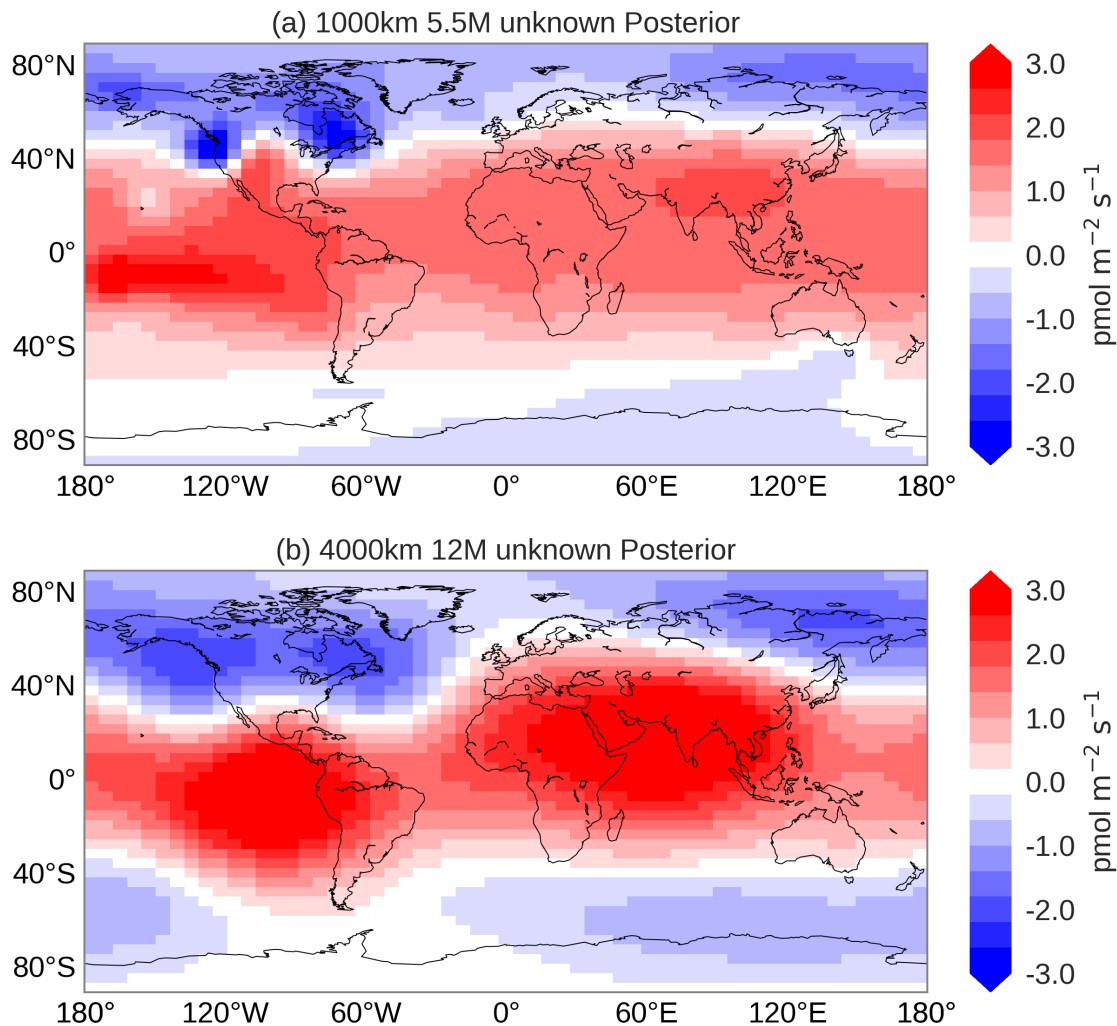

**Figure 5.** Optimized emission pattern of the "unknown" field of inversion $S_u$ for different settings of the spatio-temporal correlation lengths (a) Spatial correlation of 1000 km and temporal correlation of 5.5 months (b) Spatial correlation of 4000 km and temporal correlation of 12 months. Results are averages over 2008–2011.



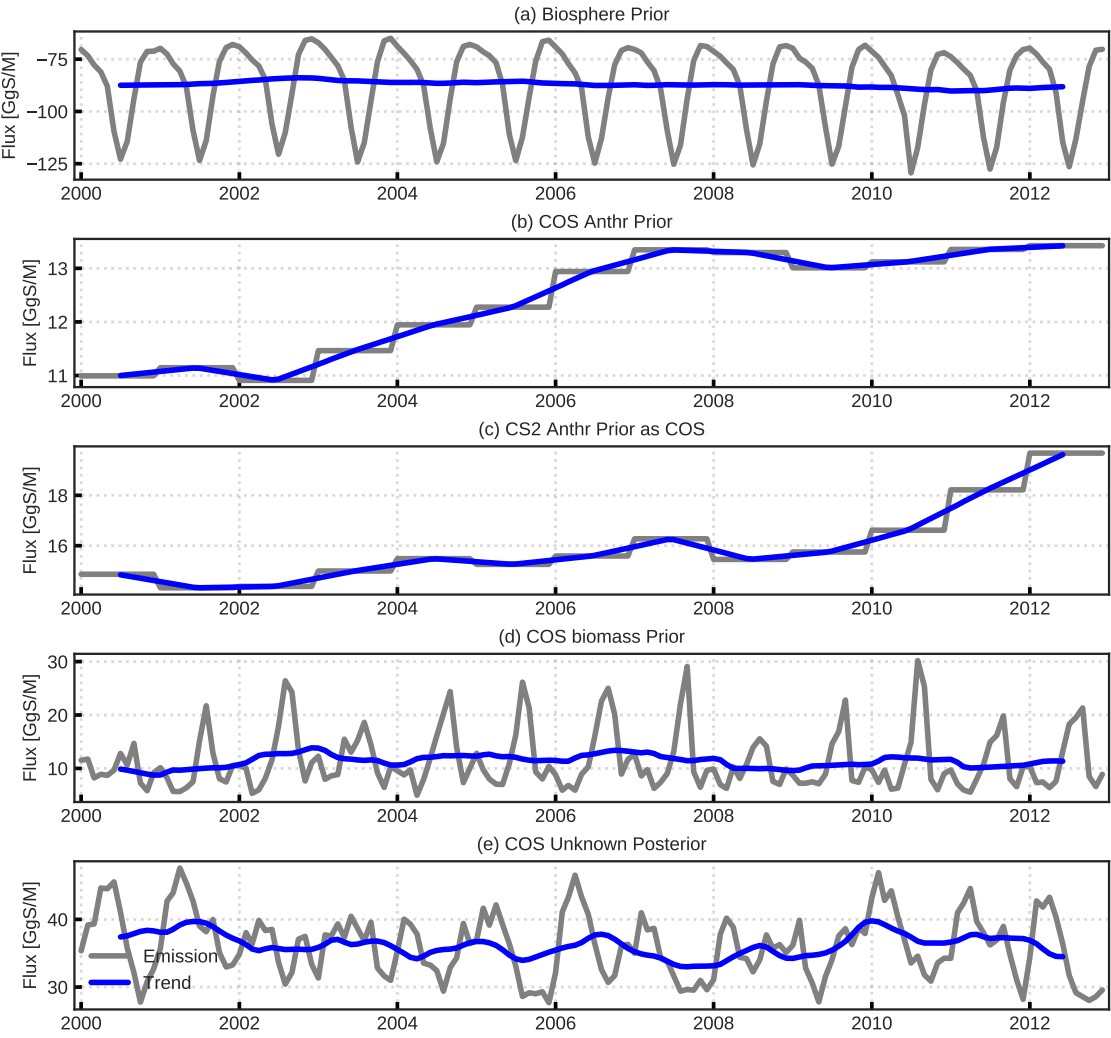

**Figure 6.** Monthly exchange fluxes related to inversion $S_u$: (a) prior biosphere (b) anthropogenic COS emissions (c) COS from anthropogenic CS$_2$ (d) biomass burning (e) optimized "unknown" emissions. Monthly global emissions (fluxes) are shown in gray, and the trend is shown in blue.



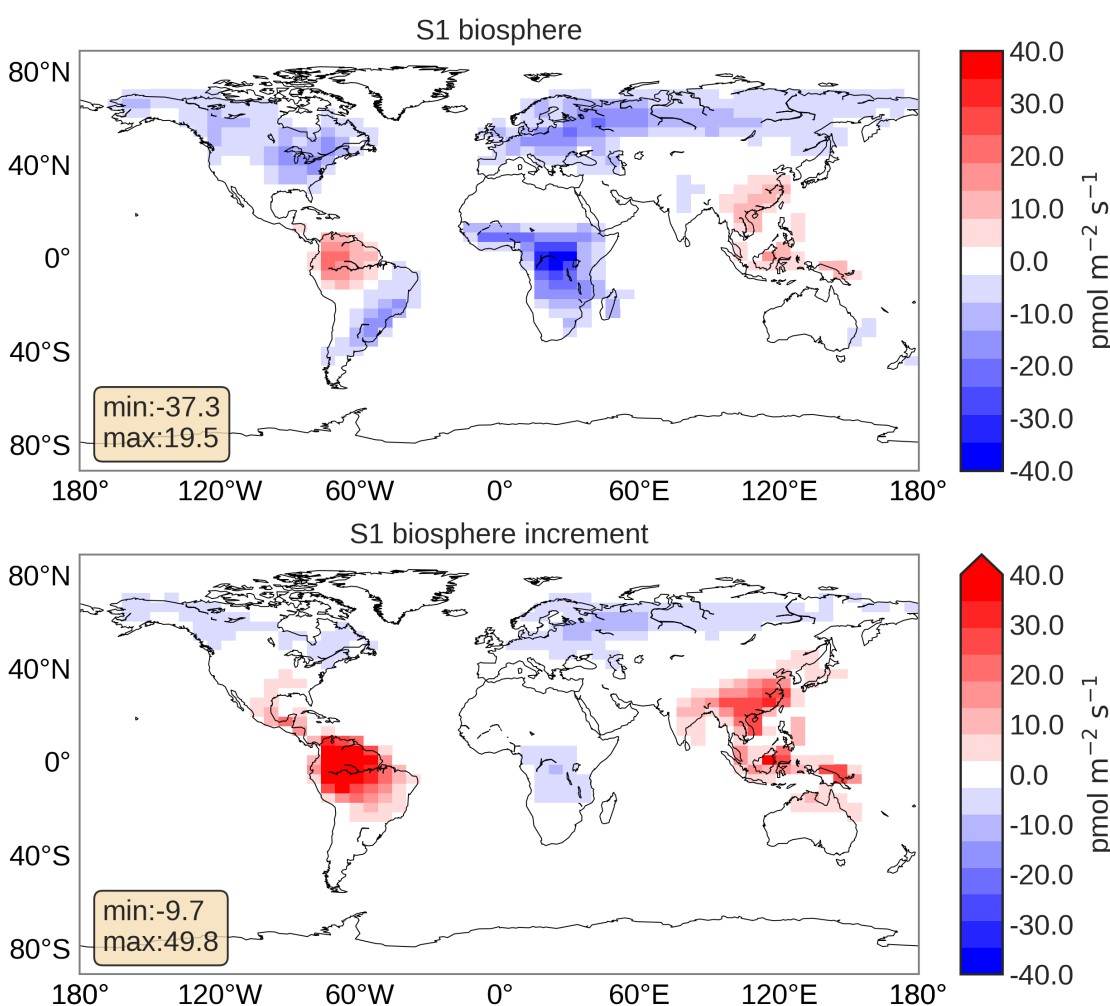

**Figure 7.** Posterior biosphere flux from inversion S1 and increment (posterior-prior). The fluxes represent 3–year (2008–2010) averages with removal of 6 month spin-up and spin-down periods. The maximum and minimum flux values are given in the boxes.



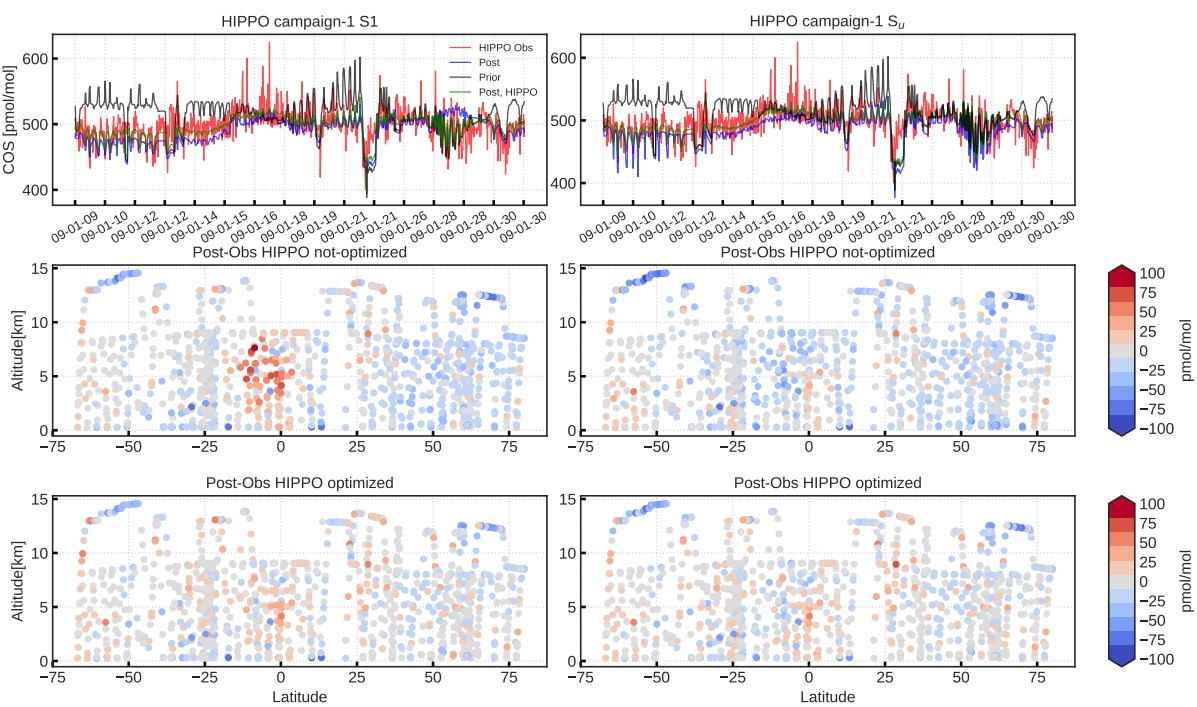

**Figure 8.** HIPPO campaign 1 COS observations compared to results from inversions S1 (left) and $S_u$ (right). The first row shows time series of HIPPO observations (red), prior (black), posterior (blue), and posterior with HIPPO observations assimilated (green). The middle and bottom rows show model minus observations in a latitude-height plot for inversions with and without assimilating HIPPO observations, respectively.

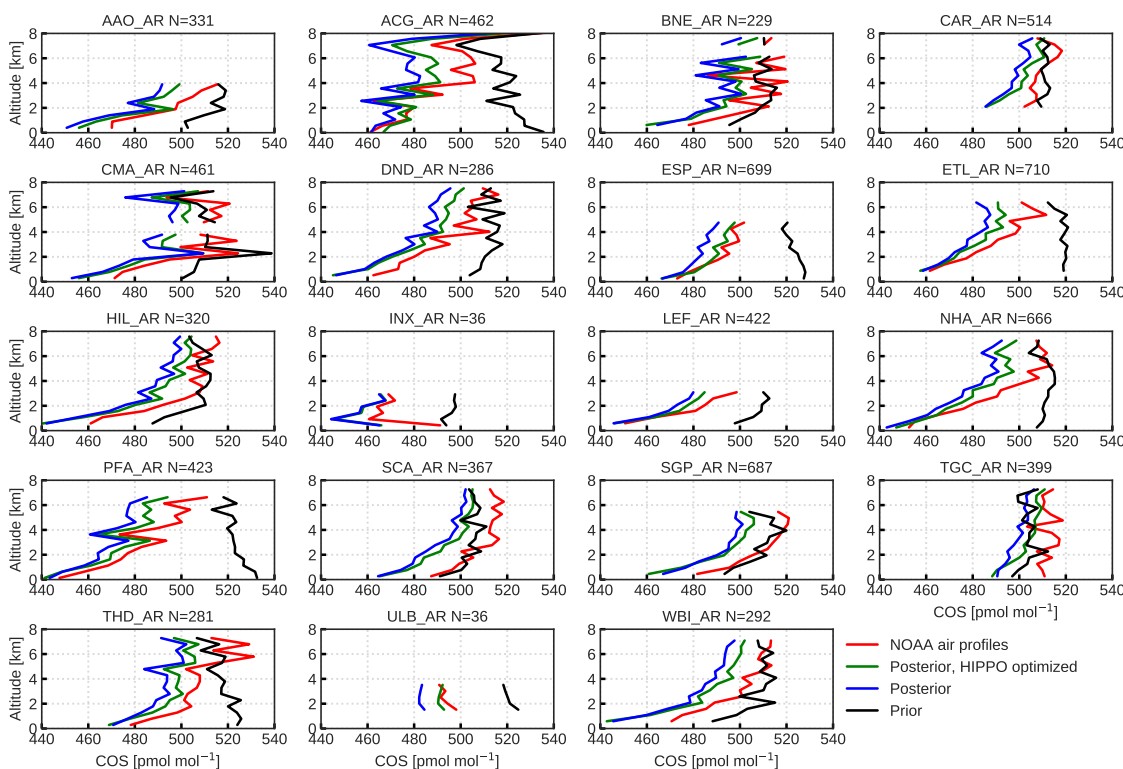

**Figure 9.** Prior (black) and posterior (blue) profiles of inversion S1, compared to NOAA aircraft profiles (red). Location (see Figure 1) and number of observations are mentioned in the caption. The green lines are results from an inversion in which, next to NOAA surface observations, also HIPPO observations are assimilated.

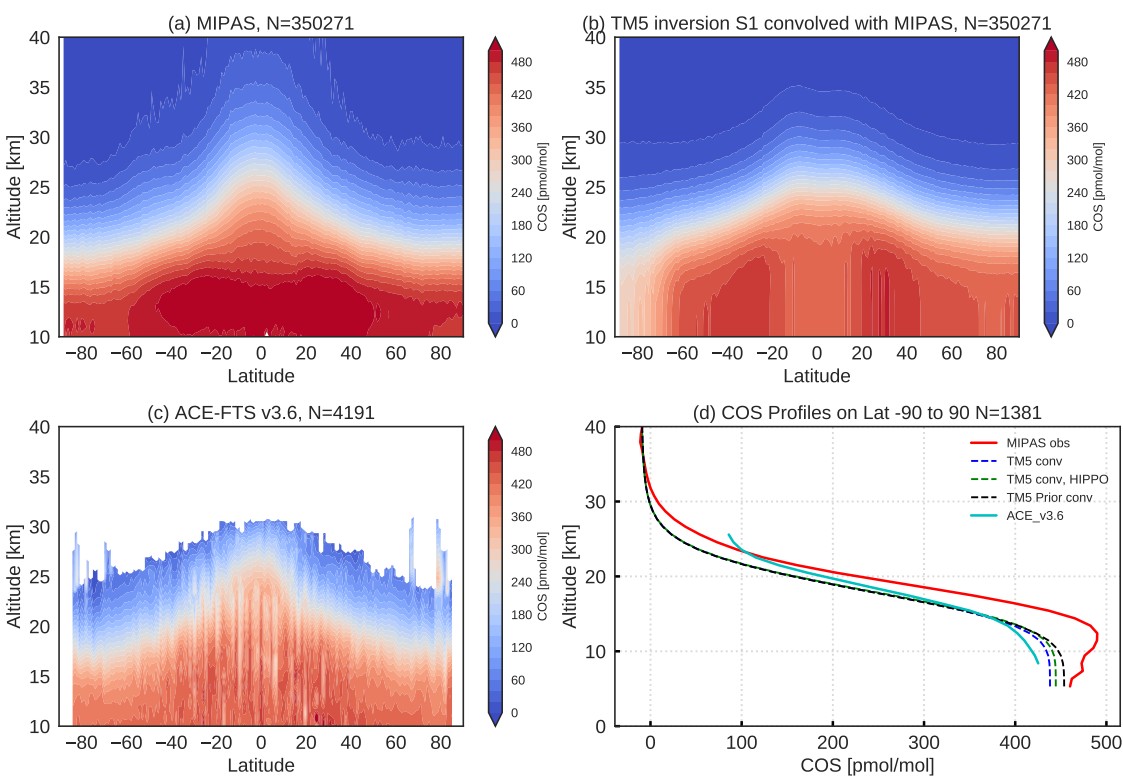

**Figure 10.** Comparison of MIPAS and ACE-FTS v3.6 with TM5 results from inversion S1 in 2009. (a) latitude–height contour plot of MIPAS; (b) TM5 S1 convolved with the MIPAS AK; (c) ACF-FTS profiles; (d) average of collocated profiles for MIPAS (red), TM5 convolved with MIPAS AK from inversion S1 (blue), TM5 convolved from inversion S1 (with HIPPO observations assimilated) (green), TM5 convolved prior (black), and ACE-FTS. In (d) TM5 and MIPAS profiles are collocated with respect to ACE-FTS profiles within a temporal offset of 6 hr and a spatial distance within 5 degrees. The number of collocated profiles is 1381.

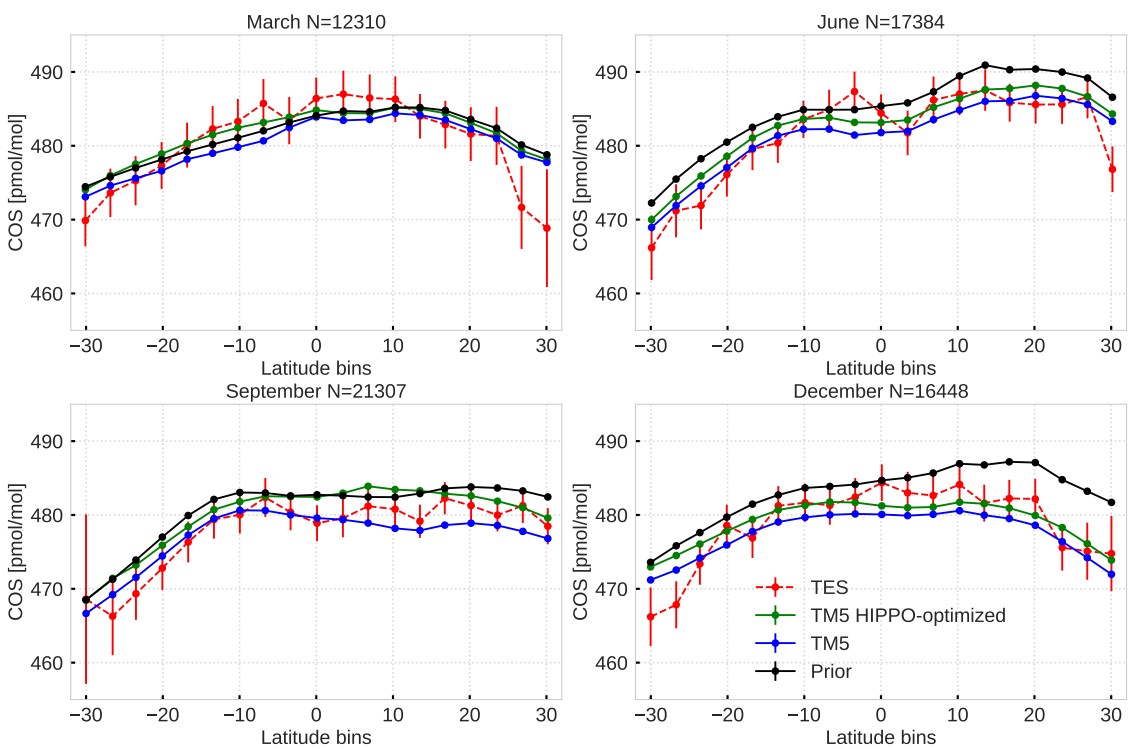

**Figure 11.** Column averaged COS mole fractions sampled by TES (red), model prior (black), model posterior (blue) and posterior with HIPPO observations assimilated (green) for March, June, September and December. The columns are averaged over 2008–2010 in 20 latitudinal bins from 32°S to 32°N. The result of inversion S1 is shown. Error bars on TES represent variability in the measurements and the number of observations is given in the caption. Variability is mostly determined by measurement noise.





**Table 1.** The split of anthropogenic emissions in the different categories and between COS and $CS_2$ based on Zumkehr et al. (2018). Note that we used a $CS_2$ to COS molar yield of 0.87 and that $CS_2$ contains two S atoms. Averages over 2000–2012 are presented.

| Emission Type | Total COS | Fraction COS[a] | Direct COS | Direct $CS_2$ |
|---|---|---|---|---|
| | Gg S a$^{-1}$ | % | Gg S a$^{-1}$ | Gg S a$^{-1}$ |
| Agricultural Chemicals | 16.9 | 0.0 | 0.0 | 38.9 |
| Aluminum Smelting | 22.2 | 88.2 | 19.6 | 6.0 |
| Industrial Coal | 52.1 | 99.5 | 51.8 | 0.7 |
| Residential Coal | 54.0 | 100.0 | 54.0 | 0.0 |
| Industrial Solvents | 5.4 | 0.0 | 0.0 | 12.5 |
| Carbon Black | 19.7 | 26.5 | 5.2 | 33.3 |
| Titanium Dioxide | 39.4 | 26.5 | 10.5 | 66.6 |
| Pulp & Paper | 0.1 | 3.2 | 0.0 | 0.3 |
| Rayon Yarn | 41.1 | 0.0 | 0.0 | 94.6 |
| Rayon Staple | 77.3 | 0.0 | 0.0 | 177.7 |
| Tires | 15.1 | 43.0 | 6.5 | 19.8 |
| Total Anthropogenic | 343.3 | - | 147.5 | 450.2 |

[a] The fraction of COS is calculated based on the COS to $CS_2$ emission ratio reported in Table 1 of Lee and Brimblecombe (2016).



**Table 2.** Biomass burning emission factors used in converting COS emissions. EF COS denotes the COS emission factor from dry mass in units g kg$^{-1}$ COS per dry mass, and EF CO denotes the CO emission factor in g kg$^{-1}$ CO per dry mass. Emission factors were taken from Andreae (2019).

| | EF COS | EF CO |
| --- | --- | --- |
| | g kg$^{-1}$ COS per dry mass | g kg$^{-1}$ CO per dry mass |
| Savanna and grassland | 0.038 | - |
| Tropical forest | 0.078 | - |
| Temperate forest | 0.035 | - |
| Boreal forest | 0.058 | - |
| Peat fires | 0.110 | - |
| Agricultural waste burning | 0.059 | - |
| Biofuel burning without dung | 0.017 | 83 |
| Biofuel burning with dung | 0.210 | 89 |





**Table 3.** Names and error settings of the inversions performed in this study. The values correspond to grid-scale errors. Monthly flux fields are optimized using spatial and temporal correlation lengths of 4000 km and 12 months, except for inversion $S_u$, in which multiple settings are explored.

|       | Biosphere | Ocean COS | Ocean CS$_2$ | Biomass burning | Anthropogenic COS and CS$_2$ | "Unknown" |
|-------|-----------|-----------|--------------|-----------------|------------------------------|-----------|
| $S_u$ | -         | -         | -            | -               | -                            | 100%      |
| S1    | 50%       | 50%       | 50%          | 10%             | 10%                          | -         |
| S2    | -         | 50%       | 50%          | -               | -                            | -         |
| S3    | 50%       | -         | -            | -               | -                            | -         |



**Table 4.** Results from inversions $S_u$, S1, S2, and S3 compared to published global COS budgets

| COS Budget (Gg S a$^{-1}$) | Kettle2002[a] | Montzka2007[b] | Berry2013[c] | Kuai2015[d] | Our prior | $S_u$ | S1 | S2 | S3 |
|---|---|---|---|---|---|---|---|---|---|
| Direct oceanic COS | 41 | 40 | 39 | 41 | 40 | 40 | -18 | 22 | 40 |
| Indirect oceanic CS$_2$ as COS | 84 | 240 | 81 | 83 | 81 | 81 | 96 | 499 | 81 |
| Indirect oceanic DMS as COS | 154 | - | 156 | 155 | 156 | 156 | 156 | 156 | 156 |
| Direct anthropogenic COS | 64 | 64 | 64 | 62 | 155 | 155 | 153 | 156 | 155 |
| Indirect anthropogenic CS$_2$ as COS | 116 | - | 116 | 113 | 188 | 188 | 188 | 189 | 188 |
| Indirect anthropogenic DMS as COS | 1 | - | 1 | 0 | 6 | 6 | 6 | 6 | 6 |
| Biomass burning | 38 | 106 | 136 | 49 | 136 | 136 | 124 | 136 | 128 |
| Additional ocean flux | - | - | 600 | 559 | - | - | - | - | - |
| Anoxic soils | 26 | 66 | - | - | - | - | - | - | - |
| *Sources* | *523* | *516* | *1193* | *1062* | *762* | *762* | *705* | *1163* | *754* |
| Destruction by OH | -94 | -96 | -101 | -111 | -101 | -101 | -103 | -101 | -101 |
| Destruction by O | -11 | -11 | - | - | - | - | - | - | - |
| Destruction by photolysis | -16 | -16 | - | - | -40 | -40 | -40 | -40 | -40 |
| Uptake by plants | -238 | -1115 | -738 | -775 | -1053 | -1053 | -557 | -1053 | -613 |
| Uptake by soil | -130 | -127 | -355 | -176 | - | - | - | - | - |
| *Sinks* | *-489* | *-1365* | *-1194* | *-1062* | *-1194* | *-1194* | *-700* | *-1194* | *-754* |
| *Unknown* | - | - | - | - | *432* | *425* | - | - | - |
| *Net total* | *34* | *-849* | *-2* | *0* | *0* | *-6* | *5* | *-31* | *0* |

[a] Kettle et al. (2002)
[b] Table 2 from Montzka et al. (2007)
[c] Berry et al. (2013)
[d] Kuai et al. (2015)





**Table 5.** $\chi^2$ metrics and mean biases for the different inversion scenarios. Statistics are shown for the NOAA surface stations, the HIPPO campaigns, and the NOAA airborne profiles. Biases are given in pmol mol$^{-1}$.

| Inversion scenario | HIPPO optimized[a] | Metric | HIPPO | NOAA surface | NOAA airborne |
|---|---|---|---|---|---|
| | No | $\chi^2$ | 40.7 | 1.9 | 26.0 |
| | No | Bias | -13.9 | 0.0 | -12.4 |
| S$_u$ | Yes | $\chi^2$ | 4.7 | 2.5 | 17.3 |
| | Yes | Bias | -1.1 | 1.5 | -8.3 |
| | No | $\chi^2$ | 43.8 | 2.4 | 27.7 |
| | No | Bias | -12.0 | -0.4 | -13.8 |
| S1 | Yes | $\chi^2$ | 4.8 | 2.9 | 20.1 |
| | Yes | Bias | -1.3 | 1.3 | -9.7 |
| | No | $\chi^2$ | 54.2 | 4.9 | 48.2 |
| | No | Bias | -19.4 | 1.5 | -16.7 |
| S2 | Yes | $\chi^2$ | 6.3 | 5.9 | 27.0 |
| | Yes | Bias | -4.6 | 7.5 | -5.9 |
| | No | $\chi^2$ | 43.3 | 2.5 | 27.5 |
| | No | Bias | -12.3 | -0.2 | -14.3 |
| S3 | Yes | $\chi^2$ | 5.0 | 3.2 | 21.1 |
| | Yes | Bias | -1.4 | 1.6 | -10.5 |

[a] If HIPPO is not optimized, only NOAA surface data is assimilated in inversions. If HIPPO is optimized, both NOAA surface data and HIPPO are assimilated in inversions. NOAA airborne data is only used for validation.