# Peer review of "Inverse modelling of carbonyl sulfide: implementation, evaluation and implications for the global budget"

_Atmospheric Chemistry and Physics, 2020_

## Referee Comment (RC1) · J. Elliott Campbell (Referee) · 29 Jul 2020

In this study, the authors present a global source/sink inversion for atmospheric OCS. The study is particularly notable for using a 4DVar approach with OCS, implementing the indirect sources of OCS as separate tracers, and applying a broad suite of observations as constraints, and considering alternative state variables for the inversion. This study advances the understanding of OCS sources and sinks using large-scale data and provides a foundation for critical next steps to implement a first-order sink inversion with satellite data.

Major Comments

To what extent can you discuss and analyze the importance of implementing CS2 as a

separate tracer? Previous previous studies assume CS2 emissions convert on emission and only have a single OCS tracer. I wonder if one region this may be important is the Asian anthropogenic outflow. In previous studies without the separate tracers, the high mixing ratios should be more immediately over Asia while in your TM5 simulations the highest mixing ratios should be somewhat downstream of the Asian source. Perhaps a feature like this can be seen in TES data. It may be relevant to draw comparisons to related studies with CO2 such as: Suntharalingam, P., Randerson, J. T., Krakauer, N., Logan, J. A. and Jacob, D. J.: Influence of reduced carbon emissions and oxidation on the distribution of atmospheric CO2: Implications for inversion analyses, Global Biogeochem. Cycles, 19, GB4003, 2005.

In the abstract, the authors find that the missing source shows little inter-annual variation but large seasonal variation. While figure 6e provides some information on this seasonality, further plots and discussion would be helpful to explore this seasonality. Consider adding maps to the supplement of the optimized fluxes from 6e for 4 seasons and time series of regional averages.

The abstract notes that an overestimated sink cannot be ruled out but the manuscript notes that tropical land constraints are not available. This note about missing tropical continental boundary layer data should be added to the abstract for clarity.

Can the authors reconsider this statement about the overestimate sink by further use of the MIPAS data? I don't think you need another inversion run but just a few plots to compare the S1 and S3 runs to the geographic variability in MIPAS. Note that the MIPAS data clearly show global minimums in the convective outflow of the Amazon? I don't think Fig10 is sufficient to explore the MIPAS constraint because the critical dimension is longitude. In the MIPAS tropics there is high mixing ratios in the western tropical Pacific and low mixing ratios over Amazon/Congo. Maps of TM5 at high altitude along with maps of MIPAS are needed to test the validity of the large changes in the biosphere flux from inversions S1 and S3.

Section 2.2.3 could use some additional explanation with respect to: "The SiB4 model was constrained by a prescribed COS mole fraction of 500 pmol mol$-1$ outside of canopy." This 500 pmol mol-1 is a placeholder. The actual boundary layer mixing ratio is lower and this SiB flux is best implemented in a model using a first-order dependency on ambient levels. Thus the 1053 GgS/y is likely an overestimate which is consistent with the lower flux reported in Berry et al of 738 GgS/y in which the first-order relationship is used. I think its fine that this study uses a zero-order approach but I think this should be carefully distinguished in the methods from future work that will need to implement the first-order approach. Furthermore it could be noted later in the abstract when the correction is made to obtain 851 GgS/y that this result is closer to the Berry et al result of 738 GgS/y.

Minor Comments

Maybe adjust the abstract wording to slightly improve clarity that plant sink in the TM5 runs are zero order (not first order). For example, "We finally find that the biosphere flux dependency on surface COS mole fraction (which was not modeled in this study) may substantially. . ."

The introduction notes Suntharalingam et al. (2008) study which attempted to fit background data by increasing the plant sink but you may also want to reference the Campbell et al. (2008) finding that this upward revision could be validated using direct observations from the continental boundary layer from the intensive INTEX-NA airborne campaign.

Campbell, J. Elliott, et al. "Photosynthetic control of atmospheric carbonyl sulfide during the growing season." Science 322.5904 (2008): 1085-1088.

The authors site Lennartz et al. (2017) for the bottom-up ocean emissions but please also note the upward revision in Lennartz et al. (2019).

Lennartz, S. T., von Hobe, M., Booge, D., Bittig, H. C., Fischer, T., Gonçalves-Araujo,

R., Ksionzek, K. B., Koch, B. P., Bracher, A., Röttgers, R., Quack, B., and Marandino, C. A.: The influence of dissolved organic matter on the marine production of carbonyl sulfide (OCS) and carbon disulfide (CS2) in the Peruvian upwelling, Ocean Sci., 15, 1071–1090, https://doi.org/10.5194/os-15-1071-2019, 2019.

Section 2.1.1 should note the lack of observatories in the tropical continental boundary layer.

Regarding section 2.2.1, how did you divide the anthropogenic emissions into direct COS and indirect CS2? Did the Zumkehr data file present emissions separately for direct and indirect? If they didn't then how did you back this out? You may want to look at the emission inventory in Campbell et al (2015) which does present separate emission estimates for direct COS and indirect CS2. The Zumkehr approach was an extension of Campbell et al 2015.

Campbell, JE, Whelan, ME, Seibt, U, Smith, SJ, Berry, JA, and Hilton, TW (2015), Atmospheric carbonyl sulfide sources from anthropogenic activity: Implications for carbon cycle constraints. Geophys. Res. Lett., 42, 3004– 3010. doi: 10.1002/2015GL063445.

Section 2.2.1 discussed uncertainties in molar yield. It might be worth noting that the uncertainty in the anthropogenic inventory is much larger than the uncertainty in molar yield.

Section 2.2.2 might want to draw comparison of this studies results to previous estimates from biomass burning in Campbell et al (2015) and open burning in Stinecipher et al. (2019).

Regarding the poor posterior fit at NWR and THD, are there references in the CO2 inversion literature that had the same difficulty? These sites are designed to capture background mixing ratio but sometimes they suffer from local influence which would be one reason for the poor posterior fit. One helpful paper you may want to reference is Riley et al.

Riley, W. J., Randerson, J. T., Foster, P. N., and Lueker, T. J. (2005), Influence of terrestrial ecosystems and topography on coastal CO2 measurements: A case study at Trinidad Head, California, J. Geophys. Res., 110, G01005, doi:10.1029/2004JG000007.

Line 421: "However, observations clearly show a large drawdown of COS near the surface (Hilton et al., 2017; Spielmann et al., 2020)." You may want to reference the INTEX-NA data which is an intensive sampling of vertical profiles in the continental boundary layer (Campbell et al.,2008)

---

## Referee Comment (RC2) · Anonymous Referee #2 · 5 Aug 2020

Dear Jin Ma and colleagues,

This study tackles the issue of closing the global OCS budget. In the past, a missing source was often assumed to be in the ocean, though actual oceanic observations are not extensive enough to answer the question well. Another possibility is that the sink is overestimated. Using TM5-4DVAR is a good method of trying to get at this question.

Comments

The uptake of OCS is tied to the OCS concentration within the canopy. There are large variations in OCS uptake as OCS depleted air flows through vegetation, e.g. Berkelhammer et al. 2020 (https://doi.org/10.1029/2019GL085652). Most canopies do not see free troposphere concentrations of OCS. This will affect the plant uptake flux

component substantially and should be addressed.

DMS should not be considered a major source of OCS. Many researchers still refer to the 7% yield figure from the Barnes 1996 paper, but note that chamber studies proceeded without NOx and at high DMS concentrations. Subsequent studies demonstrated that an alternative chemical pathway is typically taken, and that changes in NOx affect OCS formation greatly (e.g. https://doi.org/10.1016/S1352-2310(98)00120-4). In other words, if you would like to include DMS, it is important to also model NOx. This is most likely such a small contribution that in the Whelan et al., 2018 synthesis, it was concluded that DMS should only be included as a source of uncertainty in the ocean flux rather than a source itself.

If the lifetime of CS2 was 12 days, it might make sense to model CS2 separately, since the associated OCS will not show up in the air parcel until it has traveled nearly around the globe. However, more recent evidence suggested that the lifetime is much shorter than that. For example, see the 3D atmospheric transport study performed by Anwar Khan which focusses only on CS2 and estimates a lifetime of less than 4 days at maximum: 10.3934/environsci.2017.3.484.

The SiB3 model had a known phenology problem, where the growing season starts too soon (by a couple of weeks, perhaps) and ends too early. Does SiB4 have this issue? It would be good to check the seasonal timing by comparing to SiB3 or even another slightly complicating proxy, e.g. SIF.

The abstract should be revisited to better reflect the conclusions of the study and to tidy up the language, e.g. sources of OCS are obviously included in current budgets.

In short, revising the inclusion of DMS and addressing the first order plant uptake issue will certainly create a different budget overall, and may affect the conclusions.

---

## Author Comment (AC3) · 14 Dec 2020

We have an additional short response to the following comment, based on our recent analysis of the  $CS_2$  lifetime (Khan et al. 2017).

**Comments**

If the lifetime of  $CS_2$  was 12 days, it might make sense to model  $CS_2$  separately, since the associated OCS will not show up in the air parcel until it has traveled nearly around the globe. However, more recent evidence suggested that the lifetime is much shorter than that. For example, see the 3D atmospheric transport study performed by Anwar Khan which focusses only on  $CS_2$  and estimates a lifetime of less than 4 days at maximum: 10.3934/environsci.2017.3.484.

We thank the reviewer for pointing out this more recent paper regarding the  $CS_2$  lifetime. We implemented the  $CS_2$  + OH chemistry using the rate quoted in (Khan et al. 2017). As a result, we spotted a mistake in the rate constant of the  $CS_2$  + OH reaction in the Khan et al. (2017) paper. According to (Khan et al. 2017), the rate constant of  $CS_2$ +OH is (their Table 1):

rate constant 1 =
$$\frac{1.25 \times 10^{-16} e^{\frac{4550}{T}}}{(1+1.81 \times 10^{-13} e^{\frac{3400}{T}})}$$

However, the rate constant (Sander et al., 2006, Hynes et al., 1988) should read:

rate constant 2 =
$$\frac{1.25 \times 10^{-16} e^{\frac{4550}{T}}}{(T+1.81 \times 10^{-3} e^{\frac{3400}{T}})}$$

We verified with the authors of Khan et al. (2017) that this is a typo and that the rate constant is correctly implemented in their model. We also implemented the Sander et al. (2006) rate constant in our simulations. Results are shown in Figure 1.

*Figure 1: Monthly averaged burden (left) and atmospheric lifetime (right) of CS2, calculated in TM5 with the rate constant of Sander et al. (2006) implemented.*

As can be seen in Figure 1, the burden and lifetime vary considerably over the year. The yearly average lifetime (burden/loss) amounts to 6.2 days, substantially larger than the value quoted in Khan et al. (2017): 2.8-3.4 days. Possible reasons are (1) our burden is higher due to larger

emissions and possibly lower OH (2) no deposition has been implemented in our simulations. In Khan et al. (2017) loss through deposition in their standard run account for  $\sim 15\%$  of the CS2 removal. Note also that their standard model simulation underestimated CS2 mole fractions, specific over large emission regions.

For comparison, Figure 2 shows the results from our standard run, with a constant  $CS_2 + OH$  rate from Jones et al. (1982):  $2.0 \times 10^{-12}$  cm3 molecules-1 s-1.

*Figure 2: Monthly averaged burden (left) and atmospheric lifetime (right) of CS2, calculated in TM5 with the rate constant of Jones et al. (1982) implemented.*

As can be seen, the lifetime and burden are slightly larger using this rate. The yearly average lifetime amounts to 9.4 days. Figures 3 and 4 show the January and July COS mole fraction difference at the surface between our standard model and the Sander et al. (2006) rate. It can be seen that the differences are relatively small and remain smaller than 10 pmol mol-1.

Since the lifetime in TM5 is relatively long (> 10 days in the Northern Hemisphere winter), we would argue that it is necessary to simulate  $CS_2$  as a separate tracer, to model the delayed COS production from  $CS_2$  oxidation. To highlight this further, we will (in the Supplement of the revised paper) show results from a simulation in which we emit  $CS_2$  directly as COS.

Figure 3. COS mole fractions difference between this study and the rate as (Sander et al., 2006). This is the average COS mole fraction difference in January 2008-2010. The maximum and minimum values of the difference are marked in the left bottom corner.

---

## Author Comment (AC2)

**Dear reviewer,**

We appreciate your effort and time on the reviewing work, especially during the epidemic time. We have copied the remarks from reviewer #2 as below, and put our response to the comments point by point. The text in blue color is from reviewer #2 and the black text is our response.

**Anonymous Referee #2**

We would like to thank reviewer #2 for their time in reviewing this manuscript.

**Comments**

The uptake of OCS is tied to the OCS concentration within the canopy. There are large variations in OCS uptake as OCS depleted air flows through vegetation, e.g. Berkelhammer et al. 2020 (https://doi.org/10.1029/2019GL085652). Most canopies do not see free troposphere concentrations of OCS. This will affect the plant uptake flux component substantially and should be addressed.

This comment was also made by reviewer #1. Because we use an inversion framework, we employ a zero-order flux approach, which is technically easy to implement. Indeed, we find adjustments that point to a large drawdown of COS in the canopy. Like we mention in the discussion, in analyzing the flux adjustment, we should be aware that part of the adjustment is due to the "concentration" effect. Based on the comments of both reviewers, we will make this issue clearer in the revised manuscript.

DMS should not be considered a major source of OCS. Many researchers still refer to the 7% yield figure from the Barnes 1996 paper, but note that chamber studies proceeded without NOx and at high DMS concentrations. Subsequent studies demonstrated that an alternative chemical pathway is typically taken, and that changes in NOx affect OCS formation greatly (e.g. https://doi.org/10.1016/S1352-2310(98)00120-4). In other words, if you would like to include DMS, it is important to also model NOx. This is most likely such a small contribution that in the Whelan et al., 2018 synthesis, it was concluded that DMS should only be included as a source of uncertainty in the ocean flux rather than a source itself.

Currently we do not simulate NOx chemistry in the COS inverse modelling, since this largely increases the complexity of the modelling system. However, DMS is normally emitted over remote oceans, where low NOx concentrations prevail. We do not fully agree with the reviewer that the COS yield of 0.7% should not be included as long as the COS budget is not closed. Nevertheless, we will perform a NO-DMS inversion in which we place the DMS-related COS emissions to the "unknown" category.

If the lifetime of CS2 was 12 days, it might make sense to model CS2 separately, since the associated OCS will not show up in the air parcel until it has traveled nearly around the globe. However, more recent evidence suggested that the lifetime is much shorter than that. For example, see the 3D atmospheric transport study performed by Anwar Khan which focusses only on CS2 and estimates a lifetime of less than 4 days at maximum: 10.3934/environsci.2017.3.484.

We thank the reviewer to point out this publication. We have checked the rate constant used in our paper for CS2+OH, which is  $2x10^{12}$  cm3 molecules-1 s-1 (line 214). Assuming that the OH average concentration is about 106 molecules cm-3, we recalculated the lifetime as about 5.79 days. We cited the reference Khalil & Rasmussen (1984), but the rate constant is from another paper (Jones et al., 1983). The lifetime quoted in the paper will be corrected to ~6 days accordingly. Khan et al. (2017) presented a global modelling study of CS2 and quote a lifetime of 2.8-3.4 days, based on a different evaluation of the CS2+OH rate (Sander et al., 2006). This this lifetime is still half of our lifetime, and because reviewer #1 also pointed to the novelty of including the CS2 precursor, we will present a sensitivity study for the CS2 lifetime in the revised manuscript.

The SiB3 model had a known phenology problem, where the growing season starts too soon (by a couple of weeks, perhaps) and ends too early. Does SiB4 have this issue? It would be good to check the seasonal timing by comparing to SiB3 or even another slightly complicating proxy, e.g. SIF.

Our biosphere fluxes are based on simulations with the Simple Biosphere model, version 4 (SiB4) (Haynes et al., 2019). New in SiB4 compared to SiB3 (used by Berry et al. (2013)) are capabilities to simulate i.e. carbon pools, land cover heterogeneity, and leaf phenology. Further, COS uptake formulations in SiB4 are the same as in Berry et al. (2013). One of the new features in SiB4 compared to SiB3 is that it includes prognostic phenology, and this phenology is no longer depending on satellite products. So, if SiB3 had a problem with the phenology this does not automatically mean that SiB4 has the same problem. In fact, from an ongoing comparison with field observations the phenology looks good in SiB4. Results will be presented in a manuscript that is currently in preparation by co-author Linda Kooijmans. Therefore, we consider further discussion of the prior SiB4 fluxes beyond the scope of this study.

The abstract should be revisited to better reflect the conclusions of the study and to tidy up the language, e.g. sources of OCS are obviously included in current budgets.

The abstract will be revised to better show the conclusions and the cohesion of language.

In short, revising the inclusion of DMS and addressing the first order plant uptake issue will certainly create a different budget overall, and may affect the conclusions.

We will discuss these issues in the revised manuscript. We argue, however, that a zero-order approach is sufficient for a first inverse modelling study, since the fluxes are allowed to adjust to the concentration effect. Concerning DMS, we consider this issue currently unresolved, and we will include a NO-DMS inversion.

References:

- 1. Khalil, M. A. K., & Rasmussen, R. A. (1984). Global sources, lifetimes and mass balances of carbonyl sulfide (OCS) and carbon disulfide (CS2) in the earth's atmosphere. Atmospheric Environment (1967), 18(9), 1805-1813.
- 2. Jones, B. M. R., Cox, R. A., & Penkett, S. A. (1983). Atmospheric chemistry of carbon disulphide. Journal of Atmospheric Chemistry, 1(1), 65-86.
- 3. Khan, A., Razis, B., Gillespie, S., Percival, C., & Shallcross, D. (2017). Global analysis of carbon disulfide (CS2) using the 3-D chemistry transport model STOCHEM. Aims Environ. Sci, 4, 484-501.

- 4. Sander SP, Friedl RR, Golden DM, et al. (2006) Chemical kinetics and photochemical data for use in atmospheric studies. Evaluation number 15, JPL publication 06-2, Jet Propulsion Laboratory, Pasadena, CA.
- Berry, J., Wolf, A., Campbell, J. E., Baker, I., Blake, N., Blake, D., ... & Stimler, K. (2013). A coupled model of the global cycles of carbonyl sulfide and CO2: A possible new window on the carbon cycle. Journal of Geophysical Research: Biogeosciences, 118(2), 842-852.
- Haynes, K. D., Baker, I. T., Denning, A. S., Stöckli, R., Schaefer, K., Lokupitiya, E. Y., & Haynes, J. M. (2019). Representing grasslands using dynamic prognostic phenology based on biological growth stages: 1. Implementation in the Simple Biosphere Model (SiB4). Journal of Advances in Modeling Earth Systems, 11(12), 4423-4439.

---

## Author Response (AR1)

Dear Editor:

Thank you for allowing us to revise and improve our manuscript. We are thankful to reviewers for their critical comments and constructive feedbacks on our manuscript. We have done our best to incorporate all the comments into our manuscript in this revised version. We believe the reviewer's input greatly improved the quality of our manuscript. Below is our detailed responses. Reviewers' comments are in blue and our responses are in black. The differences between the submitted and revised manuscript are marked for reading convenience, and the pdf files including supplement are attached at the end of this file.

Jin Ma (On behalf of co-authors)
Email: j.ma@uu.nl

**J. Elliott Campbell (Referee)**

We would like to thank Professor Campbell for his time and effort to read our paper and evaluate our research.

Major Comments

To what extent can you discuss and analyze the importance of implementing CS2 as a separate tracer? Previous studies assume CS2 emissions convert on emission and only have a single OCS tracer. I wonder if one region this may be important is the Asian anthropogenic outflow. In previous studies without the separate tracers, the high mixing ratios should be more immediately over Asia while in your TM5 simulations the highest mixing ratios should be somewhat downstream of the Asian source. Perhaps a feature like this can be seen in TES data. It may be relevant to draw comparisons to related studies with CO2 such as: Suntharalingam, P., Randerson, J. T., Krakauer, N., Logan, J. A. and Jacob, D. J.: Influence of reduced carbon emissions and oxidation on the distribution of atmospheric CO2: Implications for inversion analyses, Global Biogeochem. Cycles, 19, GB4003, 2005.

The inclusion of CS2 in our model gives the possibility to validate the geophysical distribution and vertical profiles in the future. It is also worth to note that CS2 has a different source distribution compared to COS. E.g., CS2 direct emissions are mainly from the industrial production of rayon. Another issue is the atmospheric lifetime of CS2. As referee #2 also discussed, the lifetime of CS2 is more likely shorter than what we used in the current implementation (Kahn et al., 2017). It is more likely less than 4 days, and we will test the sensitivity of the CS2 lifetime in TM5. In the revised manuscript, we will include CS2 sensitivity tests by varying the lifetime from ~6 days (current setting), to 3 days (Kahn et al., 2017). Another sensitivity test that we will report on is the instantaneous transfer of CS2 to COS, and compare it with the current model settings.

In the abstract, the authors find that the missing source shows little inter-annual variation but large seasonal variation. While figure 6e provides some information on this seasonality, further plots and discussion would be helpful to explore this seasonality. Consider adding maps to the supplement of the optimized fluxes from 6e for 4 seasons and time series of regional averages.

Thanks for the suggestion. In order to better investigate the inter-annual variations of the missing source, we have done further time-series analysis for regions. Globally, the COS unknown posterior flux is shown, together with the trend, the seasonal signal and the residual (Figure 1). It can be seen that, globally, the seasonal signal is highest during NH spring and lowest during NH fall. The global flux was then split into 8 regions (Figure 2), and the regional COS unknown flux analyzed for these regions is shown in Figure 3. The region NH1 (North America plus part of Pacific and Atlantic Oceans, orange) shows a negative "unknown" flux, indicating that more sinks are needed. This likely points to an underestimation of the biosphere uptake in the prior, since this region (that is well constrained by observations) depicts a clear seasonal cycle in the optimized "unknown" flux. NH2 (Europe, green) and NH3 (Asia, red) have almost the same trend and seasonality, perhaps because they are difficult to separate using the available observations. Tropical regions TR0-3 have similar trend and seasonality, and generally shows a positive flux signal, with little seasonal cycle. This could represent and oceanic signal (underestimated emissions of COS or COS precursors in the prior), a signal

from biomass burning, or an overestimated biosphere sink. The ocean-dominated region SH (blue) has a near neutral flux, with a seasonal cycle that shows higher emissions in local fall and early winter.

We will add some of these analyses in the supplement or in the main text of the revised manuscript.

[Figure]

Figure 1. COS posterior flux of the optimized category "unknown" in inversion Su, with its seasonal signal and residual for the years 2000—2012.

[Figure]

Figure 2. The regional map for analysis of COS fluxes. NH1-3 are areas in Northern Hemisphere. TR0-3 are regions in Tropics. SH refers to the Southern Hemisphere.

[Figure]

[Figure]

[Figure]

Figure 3. The regional signals of the optimized COS posterior flux (inversion Su). The upper panel shows flux contribution in each region, and the middle panel shows the trend by filtering out the seasonal and residual signals. The lower panel is the mean seasonal cycle of the COS posterior flux in each region. The colors of the lines are the same as in Figure 2.

The abstract notes that an overestimated sink cannot be ruled out but the manuscript notes that tropical land constraints are not available. This note about missing tropical continental boundary layer data should be added to the abstract for clarity.

We will add this message to the abstract.

Can the authors reconsider this statement about the overestimate sink by further use of the MIPAS data? I don't think you need another inversion run but just a few plots to compare the S1 and S3 runs to the geographic variability in MIPAS. Note that the MIPAS data clearly show global minimums in the convective outflow of the Amazon? I don't think Fig10 is sufficient to explore the MIPAS constraint because the critical dimension is longitude. In the MIPAS tropics there is high mixing ratios in the western tropical Pacific and low mixing ratios over Amazon/Congo. Maps of TM5 at high altitude along with maps of MIPAS are needed to test the validity of the large changes in the biosphere flux from inversions S1 and S3.

This is a good suggestion. We indeed have the feeling that inversion S3 projects the total missing source on the biosphere, and this should impact the longitudinal MIPAS comparison. We will consider some extra figures in the Supplement.

Section 2.2.3 could use some additional explanation with respect to: "The SiB4 model was constrained by a prescribed COS mole fraction of 500 pmol mol$-1$ outside of canopy." This 500 pmol mol-1 is a placeholder. The actual boundary layer mixing ratio is lower and this SiB flux is best implemented in a model using a first-order dependency on ambient levels. Thus the 1053 GgS/y is likely an overestimate which is consistent with the lower flux reported in Berry et al of 738 GgS/y in which the first-order relationship is used. I think its fine that this study uses a zero-order approach but I think this should be carefully distinguished in the methods from future work that will need to implement the first-order approach. Furthermore it could be noted later in the abstract when the correction is made to obtain 851 GgS/y that this result is closer to the Berry et al result of 738 GgS/y.

We realize that the SIB4 model uses 500 ppt merely as a placeholder, leading to a large uptake. We agree that our current manuscript presents this too much as a deficiency of the SIB4 model, and will adjust this in the discussion, section 2.2.3, and the abstract. The preferred (but more difficult) implementation of a first-order removal in the 4DVAR framework will be addressed in a future study.

Minor Comments

Maybe adjust the abstract wording to slightly improve clarity that plant sink in the TM5 runs are zero order (not first order). For example, "We finally find that the biosphere flux dependency on surface COS mole fraction (which was not modeled in this study) may substantially. . ."

This will be fixed in the manuscript.

The introduction notes Suntharalingam et al. (2008) study which attempted to fit background data by increasing the plant sink but you may also want to reference the Campbell et al. (2008) finding that this upward revision could be validated using direct observations from the continental boundary layer from the intensive INTEX-NA airborne campaign. Campbell, J.

Elliott, et al. "Photosynthetic control of atmospheric carbonyl sulfide during the growing season." Science 322.5904 (2008): 1085-1088.

We thank the reviewer to point out the publication. This will be included in the manuscript.

The authors site Lennartz et al. (2017) for the bottom-up ocean emissions but please also note the upward revision in Lennartz et al. (2019). Lennartz, S. T., von Hobe, M., Booge, D., Bittig, H. C., Fischer, T., Gonçalves-Araujo, R., Ksionzek, K. B., Koch, B. P., Bracher, A., Röttgers, R., Quack, B., and Marandino, C. A.: The influence of dissolved organic matter on the marine production of carbonyl sulfide (OCS) and carbon disulfide (CS2) in the Peruvian upwelling, Ocean Sci., 15, 1071–1090, https://doi.org/10.5194/os-15-1071-2019, 2019.

We thank the reviewer to point out the publication, which will be included in the revised manuscript.

Section 2.1.1 should note the lack of observatories in the tropical continental boundary layer. Regarding section 2.2.1, how did you divide the anthropogenic emissions into direct COS and indirect CS2? Did the Zumkehr data file present emissions separately for direct and indirect? If they didn't then how did you back this out? You may want to look at the emission inventory in Campbell et al (2015) which does present separate emission estimates for direct COS and indirect CS2. The Zumkehr approach was an extension of Campbell et al 2015.
Campbell, JE, Whelan, ME, Seibt, U, Smith, SJ, Berry, JA, and Hilton, TW (2015), Atmospheric carbonyl sulfide sources from anthropogenic activity: Implications for carbon cycle constraints. Geophys. Res. Lett., 42, 3004– 3010. doi: 10.1002/2015GL063445.

We implemented anthropogenic emissions largely based on Zumhehr et al. (2018) and Campbell et al 2015. We separated the COS emission from the Zumkehr et al. (2018) work according to Table 1 in Lee and Brimblecombe (2016). In Table 1 the authors reported a detailed emission budget for COS and CS2 in anthropogenic categories. Then we used the ratio of this budget to roughly estimate the direct and indirect COS anthropogenic emissions. In this way we were able to separate the COS and CS2 direct anthropogenic emissions, which should be correct within the uncertainties. We will outline the followed procedure more clearly in the revised manuscript.

Section 2.2.1 discussed uncertainties in molar yield. It might be worth noting that the uncertainty in the anthropogenic inventory is much larger than the uncertainty in molar yield.

We will clarify this better in the revised manuscript. With more measurements available, we could try to reduce these uncertainties using our inverse modelling framework.

Section 2.2.2 might want to draw comparison of this studies results to previous estimates from biomass burning in Campbell et al (2015) and open burning in Stinecipher et al. (2019).

We will mention a few differences in the revised manuscript. The major difference is that we considered new biofuel emission factors in South Asia.

Regarding the poor posterior fit at NWR and THD, are there references in the CO2 inversion literature that had the same difficulty? These sites are designed to capture background mixing ratio but sometimes they suffer from local influence which would be one reason for the poor posterior fit. One helpful paper you may want to reference is Riley et al.
Riley, W. J., Randerson, J. T., Foster, P. N., and Lueker, T. J. (2005), Influence of terrestrial ecosystems and topography on coastal CO2 measurements: A case study at Trinidad Head, California, J. Geophys. Res., 110, G01005, doi:10.1029/2004JG000007.

We thank the reviewer to point out the CO2 paper concerning THD. It is true that THD as discussed in the paper of Riley et al. (2005) is more affected by the local coastal effect and the diurnal cycle of CO2 fluxes. The current TM5 inverse modelling effort applies a coarse resolution of $6° × 4°$ globally, and thus the effect of costal meteorology at THD is not well captured. Another point is that biosphere flux of COS is applied on a monthly-average basis, and does not account for a diurnal cycle. The NWR station is probably also affected by local land effects that are not well resolved in the coarse simulations.

To compare with CO2 inversions, we have investigated CarbonTracker North America data and found similarities between COS and CO2 inversions. For example, at THD, CO2 has relatively large model-data mismatches of ~11 ppm in Summer (Figure 4). For NWR, the CO2 model-data mispatch is about 3.37 ppm in Summer (Figure 5). In comparison, at MHD the CO2 model-data mismatch is only 2.04 ppm in Summer inferred from Flask observations. Note that CarbonTracker North America employs a resolution of $1° × 1°$ degree, compared to $6° × 4°$ degree in our COS study.

[Figure]

Figure 4. CO2 observation and simulation by CarbonTracker at THD (source: https://www.esrl.noaa.gov/gmd/ccgg/carbontracker/co2tser.php?ds=co2_thd_surface-flask_1_representative&ed=assim&lastds=co2_nwr_surface-flask_1_representative).

[Figure]

Figure 5. CO2 observation and simulation by CarbonTracker at NWR (source: https://www.esrl.noaa.gov/gmd/ccgg/carbontracker/co2tser.php?ds=co2_nwr_surface-pfp_1_allvalid-3magl&ed=assim&lastds=co2_nwr_surface-insitu_3_nonlocal).

Line 421: "However, observations clearly show a large drawdown of COS near the surface (Hilton et al., 2017; Spielmann et al., 2020)." You may want to reference the INTEX-NA data which is an intensive sampling of vertical profiles in the continental boundary layer (Campbell et al.,2008)

We will include the reference.

including the CS2 precursor, we will present a sensitivity study for the CS2 lifetime in the revised manuscript.

The SiB3 model had a known phenology problem, where the growing season starts too soon (by a couple of weeks, perhaps) and ends too early. Does SiB4 have this issue? It would be good to check the seasonal timing by comparing to SiB3 or even another slightly complicating proxy, e.g. SIF.

Our biosphere fluxes are based on simulations with the Simple Biosphere model, version 4 (SiB4) (Haynes et al., 2019). New in SiB4 compared to SiB3 (used by Berry et al. (2013)) are capabilities to simulate i.e. carbon pools, land cover heterogeneity, and leaf phenology. Further, COS uptake formulations in SiB4 are the same as in Berry et al. (2013). One of the new features in SiB4 compared to SiB3 is that it includes prognostic phenology, and this phenology is no longer depending on satellite products. So, if SiB3 had a problem with the phenology this does not automatically mean that SiB4 has the same problem. In fact, from an ongoing comparison with field observations the phenology looks good in SiB4. Results will be presented in a manuscript that is currently in preparation by co-author Linda Kooijmans. Therefore, we consider further discussion of the prior SiB4 fluxes beyond the scope of this study.

The abstract should be revisited to better reflect the conclusions of the study and to tidy up the language, e.g. sources of OCS are obviously included in current budgets.

The abstract will be revised to better show the conclusions and the cohesion of language.

In short, revising the inclusion of DMS and addressing the first order plant uptake issue will certainly create a different budget overall, and may affect the conclusions.

We will discuss these issues in the revised manuscript. We argue, however, that a zero-order approach is sufficient for a first inverse modelling study, since the fluxes are allowed to adjust to the concentration effect. Concerning DMS, we consider this issue currently unresolved, and we will include a NO-DMS inversion.

[Figure]

*Figure 1: Monthly averaged burden (left) and atmospheric lifetime (right) of CS2, calculated in TM5 with the rate constant of Sander et al. (2006) implemented.*

As can be seen in Figure 1, the burden and lifetime vary considerably over the year. The yearly average lifetime (burden/loss) amounts to 6.2 days, substantially larger than the value quoted in Khan et al. (2017): 2.8-3.4 days. Possible reasons are (1) our burden is higher due to larger

emissions and possibly lower OH (2) no deposition has been implemented in our simulations. In Khan et al. (2017) loss through deposition in their standard run account for ~15% of the $CS_2$ removal. Note also that their standard model simulation underestimated $CS_2$ mole fractions, specific over large emission regions.

For comparison, Figure 2 shows the results from our standard run, with a constant $CS_2$ + OH rate from Jones et al. (1982): $2.0 \times 10^{-12}$ cm$^3$ molecules$^{-1}$ s$^{-1}$.

[Figure]

*Figure 2: Monthly averaged burden (left) and atmospheric lifetime (right) of CS2, calculated in TM5 with the rate constant of Jones et al. (1982) implemented.*

As can be seen, the lifetime and burden are slightly larger using this rate. The yearly average lifetime amounts to 9.4 days. Figures 3 and 4 show the January and July COS mole fraction difference at the surface between our standard model and the Sander et al. (2006) rate. It can be seen that the differences are relatively small and remain smaller than 10 pmol mol$^{-1}$.

Since the lifetime in TM5 is relatively long (> 10 days in the Northern Hemisphere winter), we would argue that it is necessary to simulate $CS_2$ as a separate tracer, to model the delayed COS production from $CS_2$ oxidation. To highlight this further, we will (in the Supplement of the revised paper) show results from a simulation in which we emit $CS_2$ directly as COS.

[Figure]

*Figure 3. COS mole fractions difference between this study and the rate as (Sander et al., 2006). This is the average COS mole fraction difference in January 2008-2010. The maximum and minimum values of the difference are marked in the left bottom corner.*

[Figure]

*Figure 4. The same as Figure 3 but for July 2008-2010.*

[revised manuscript text omitted]

---

## Author Response (AR3)

**Response to comments on "Inverse modelling of carbonyl sulfide: implementation, evaluation and implications for the global budget" By Jin Ma et al. (2020)**
* * *
Dear Editor:

Thank you for allowing us to revise and improve our manuscript, especially during the epidemic time. The authors would like to thank the two reviewers for their critical comments and constructive feedbacks on our manuscript. We have done our best to incorporate all the comments into the revised manuscript. We believe the reviewers input greatly improved the quality of our manuscript. Below are our detailed responses. Reviewers' comments are written in blue, our responses are in **black**, and modifications made in the manuscript are in red with underscore. A list of changes is added to indicate what we have changed throughout the manuscript and the supplementary material. The differences between the submitted and revised manuscript are marked for reading convenience. We took the advantage of two sets of comments and made several major updates of our study as following:

- We provide now a regional analysis of optimized fluxes based on seasonal decomposition method
- We included the following sensitivity simulations and inversions:
  - $CS_2$ lifetime variations according to Khan et al. (2017)
  - $CS_2$ emitted directly as COS
  - An inversion without DMS as COS precursor (NO-DMS)
- We compare COS upper troposphere distributions of MIPAS and TM5 inversions

We also took the into account the editors' comments to keep the clarity of physical units. E.g., the unit of COS budget Gg S $a^{-1}$ is changed to Gg $a^{-1}$ (as S equivalents) to avoid misunderstanding.

**A list of changes made in the revised manuscript and supplement:**

- We have included more references related to this study: (Berkelhammer et al., 2020), (Lennartz et al., 2019), (Riley et al., 2005), (Campbell et al., 2008), (Jones et al., 1983), (Khan et al., 2017), (Sander et al, 2006), and (Haynes et al., 2019).
- The unit of COS flux in the manuscript was modified from Gg S $a^{-1}$ to COS flux (as S equivalents) Gg $a^{-1}$, because all fluxes in terms of mass are accounted for in sulfur mass. A footnote was added in Section 1 (introduction) where the unit Gg $a^{-1}$ was first used to clarify the unit used in this study.
- Abstract was modified to better reflect the overall information of this study.
- More information was added in Section 2.1.1 and Section 2.1.2 to better describe measurement data used in the study.
- Description of section 2.1.3 of satellite data was improved.
- English clearness and cohesion are improved throughout the manuscript.
- A discussion about the $CS_2$ lifetime was added in Section 2.3.
- We rewrote the information reflected from the regional analysis of the optimized fluxed in inversion *Su* in Section 3.1.
- We added a comparison of MIPAS data with TM5 inversions on latitudinal-longitudinal space in Section 3.4.
- We added a discussion of the importance to include $CS_2$ and DMS in this study.
- The conclusion was modified to better reflect the results of this study.
- We extended the acknowledgement to credit several contributors to this work.
- A list of modifications of Figures in the manuscript and supplements:
  - Figure 6 was replaced by a regional analysis of the optimized flux from inversion *Su*.

- We moved old Figure 6 to Supplement Figure S7.
- We added Figure 11 to show MIPAS compared to the TM5 inversions on 250 hPa during JJA.
- We added Supplement Figures S8-9 to show MIPAS compared to the TM5 inversions on 250 hPa during SON and on 150 hPa during JJA, respectively.
- We added Supplement Figure S10 to show sensitivity tests of $CS_2$ emitted directly as COS.
- We added Supplement Figure S11 to show a sensitivity test of a NO-DMS inversion.
- The unit of the COS budget was changed from Gg S $a^{-1}$ to Gg $a^{-1}$ (as S equivalent) in Figure 2, Figure S7 and Figure S12.
  - Typos were fixed throughout the manuscript.
  - NOx was modified to $NO_x$ in the manuscript.
  - mixing ratio was modified to mole fraction in the manuscript.
  - We also fully considered all co-authors' comments during the stage of discussion and revised the manuscript accordingly.

Jin Ma (On behalf of co-authors)
Email: j.ma@uu.nl
* * *
**Response to Reviewer #1**

**J. Elliott Campbell (Referee)**
Major Comments

To what extent can you discuss and analyze the importance of implementing CS2 as a separate tracer? Previous studies assume CS2 emissions convert on emission and only have a single OCS tracer. I wonder if one region this may be important is the Asian anthropogenic outflow. In previous studies without the separate tracers, the high mixing ratios should be more immediately over Asia while in your TM5 simulations the highest mixing ratios should be somewhat downstream of the Asian source. Perhaps a feature like this can be seen in TES data.

The inclusion of $CS_2$ in our model gives the possibility to validate the geophysical distribution and vertical profiles in the future. It is also worth to note that $CS_2$ has a different source distribution compared to COS. E.g., $CS_2$ direct emissions are mainly from the industrial production of rayon. To test the $CS_2$ as a single tracer or not, we have performed a sensitivity experiment in which $CS_2$ is emitted directly as COS, and compared it with the standard model used in the study. We have incorporated these analyses on Page 15 Line 463-468 of the revised manuscript:

"We tested the effect of emitting $CS_2$ ocean and anthropogenic sources directly as COS in an additional forward model simulation. As shown in Figure S10, COS mole fractions would become significantly larger close to $CS_2$ emission hot spots in Asia, Europe and the US. At selected stations (LEF in the US and MHD in Europe, as shown in Figure S10 a and b, we observe COS mole fractions that are up to 40 pmol $mol^{-1}$ higher during events where emitted $CS_2$ is advected to the station."

We also added a figure in the Supplement (S10, reproduced as A1 below).

[Figure]

*Figure A1. Sensitivity test in which CS₂ is emitted as COS compared to the standard model: (a) COS mole fractions sampled at two stations (LEF and MHD) (b) COS difference sampled at two stations (LEF and MHD) (c) COS spatial difference at the surface in January (d) COS spatial difference at the surface in July. Note that the results are from forward model simulations with prior fluxes in 2008–2010. For clarity, in panels a and b only results in 2009 are shown.*

It may be relevant to draw comparisons to related studies with CO2 such as: Suntharalingam, P., Randerson, J. T., Krakauer, N., Logan, J. A. and Jacob, D. J.: Influence of reduced carbon emissions and oxidation on the distribution of atmospheric CO2: Implications for inversion analyses, Global Biogeochem. Cycles, 19, GB4003, 2005.

We thank the reviewer for pointing out the related study of Suntharalingam et al. (2005), which investigated reduced carbon emissions and its influence on $CO_2$ flux estimation from inversion analysis using GEOS-CHEM model. However, we did not find direct relevance to our study with a focus on COS inverse modelling, therefore it is not referred to in this study.

In the abstract, the authors find that the missing source shows little inter-annual variation but large seasonal variation. While figure 6e provides some information on this seasonality, further plots and discussion would be helpful to explore this seasonality. Consider adding maps to the supplement of the optimized fluxes from 6e for 4 seasons and time series of regional averages.

Thanks for the suggestion. In order to better investigate the inter-annual variations of the missing source, we have done further time-series analysis for regions by using the seasonal decomposition analysis. We have added the regional analysis on Page 12 Lines 338-351 in the revised manuscript as following and replaced old Figure 6 by Figure A2 in the manuscript: "The global flux was subsequently split into 8 regions, and the regional COS *Su* fluxes analyzed for these regions are shown in (Figure 6). Region NH1 (North America plus part of Pacific and Atlantic Oceans, orange) shows a negative "unknown" flux, indicating that more sinks are needed. This likely points to an underestimation of the biosphere uptake in the prior, since this region (that is well constrained by observations) depicts a clear seasonal cycle in the optimized "unknown" flux. A larger sink is also needed in NH2 (Europe, green) and NH3 (Asia, red), but of smaller magnitude than NH1. Tropical regions TR0-TR3 have similar trend and seasonality, and generally show a positive flux signal, with little seasonal cycle. This could represent an oceanic signal (underestimated emissions of COS or COS precursors in the prior), a signal from biomass burning, or an overestimated biosphere sink. The ocean-dominated region SH (blue) has a near neutral flux, with a seasonal cycle that shows higher emissions in local fall and early winter."

[Figure]

*Figure A2. Regional analysis of multi-year optimized COS fluxes of inversion Su: (a) posterior flux per region (b) regions over which the posterior flux is analysed (c) trend in the decomposed signal (d) seasonal signal in the decomposed signal. Note that region colors in (b) are used in panels (a), (c) and (d).*

The abstract notes that an overestimated sink cannot be ruled out but the manuscript notes that tropical land constraints are not available. This note about missing tropical continental boundary layer data should be added to the abstract for clarity.

We have added this message to the abstract for clarity. The abstract was modified as follows on Page 1 Lines 13-14:
 "We found that the missing sources are likely located in the tropical regions, and an overestimated biospheric sink in the tropics cannot be ruled out due to missing observations in the tropical continental boundary layer".

Can the authors reconsider this statement about the overestimate sink by further use of the MIPAS data? I don't think you need another inversion run but just a few plots to compare the S1 and S3 runs to the geographic variability in MIPAS. Note that the MIPAS data clearly show global minimums in the convective outflow of the Amazon? I don't think Fig10 is sufficient to explore the MIPAS constraint because the critical dimension is longitude. In the MIPAS tropics there is high mixing ratios in the western tropical Pacific and low mixing ratios over Amazon/Congo. Maps of TM5 at high altitude along with maps of MIPAS are needed to test the validity of the large changes in the biosphere flux from inversions S1 and S3.

This is a good suggestion. We indeed have the feeling that inversion S3 projects the total missing source on the biosphere, and this should impact the longitudinal MIPAS comparison. We have reevaluated MIPAS vs TM5 inversions on latitudinal-longitudinal space on several pressure levels. The main result we have obtained is that TM5 inversions are well reflecting the COS distribution compared with MIPAS on pressure levels 250 hPa and 150 hPa during JJA and SON.

We have modified Section 3.4 Page 14 Lines 422-433 as follows:
"To compare the different inversions with respect to the simulated latitude--longitude distribution, Figure 11 shows a comparison of COS between TM5 inversions and MIPAS on 250 hPa in June to August. Similar results on 250 hPa from September to November and on

150 hPa from June to August are shown in supplement (Figure S8 and S9). MIPAS COS represents a 2002-2011 average taken from Glatthor et al. (2017). TM5 results have been averaged over 2008-2010. The distributions of COS in all inversions match relatively well with MIPAS. Note, however, that we adjusted the TM5 results by +25 pmol mol$^{-1}$ to match the colorscale of MIPAS. The COS distribution from the prior simulation correctly simulates low COS over the Amazon and Africa, but is clearly too high over Northern latitudes. This latter aspect is partly solved by the inversions. If we concentrate on the observed COS minimum over the Atlantic, Africa and the Amazon, inversions S1 and S3 shift this minimum to the east, consistent with the COS biosphere flux increment shown in Figure 7 for S1. Inversions *Su* and S2 exhibit a better comparison with MIPAS, suggesting that the large increments of the tropical biosphere over South America (Figure 7) are unrealistic. However, assigning the missing tropical source totally to ocean emissions (S2) appears to overestimate the COS drawdown over the Amazon.”

Figure A3 was added in the manuscript as Figure 11. Additional figures on 250 hPa during SON, and on 150 hPa during JJA were added in the supplement as Figures S8 and S9, respectively.

Conclusions have been modified accordingly on Page 17 Line 515-519 as:
“Comparison between TM5 inversions and satellite data shows that COS in the model is systematically lower than MIPAS or TES, and inversions reproduced the tropospheric COS spatial distribution well, specifically for inversions *Su* and S2. These comparisons indicate that the missing tropical source likely originates from a combination of underestimated ocean emissions, and overestimated biosphere uptake. Part of the tropical sources can be explained by the dependence of COS uptake on atmospheric mole fractions.”

[Figure]

Figure A3. COS mole fraction comparison of MIPAS and TM5 inversions on 250 hPa in June to August. (a-e) are TM5 prior, and inversions Su, S1, S2, S3, respectively. (f) is captured from Figure 11 in (Glatthor et al., 2017). TM5 results represent a 2008-2010 average, and MIPAS is averaged over 2002--2011. Because TM5 results are systematically lower than MIPAS, 25 pmol mol$^{-1}$ is added to the TM5 results for a better visual comparison.

Section 2.2.3 could use some additional explanation with respect to: "The SiB4 model was constrained by a prescribed COS mole fraction of 500 pmol mol−1 outside of canopy." This 500 pmol mol-1 is a placeholder. The actual boundary layer mixing ratio is lower and this SiB flux is best implemented in a model using a first-order dependency on ambient levels. Thus the 1053 GgS/y is likely an overestimate which is consistent with the lower flux reported in Berry et al of 738 GgS/y in which the first-order relationship is used. I think its fine that this study uses a zero-order approach but I think this should be carefully distinguished in the methods from future work that will need to implement the first-order approach. Furthermore it could be noted later in the abstract when the correction is made to obtain 851 GgS/y that this result is closer to the Berry et al result of 738 GgS/y.

We realize that the SIB4 model uses 500 pmol mol$^{-1}$ merely as a placeholder, leading to a large uptake. We agree that our current manuscript presents this too much as a deficiency of the SIB4 model, and have clarified this in the abstract and further discussed in the section 2.2.3. The

preferred (but more difficult) implementation of a first-order removal in the 4DVAR framework will be addressed in a future study.

Section 2.2.3 Page 7 Lines 183-184 was modified as:
"This 500 pmol mol$^{-1}$ is merely as a placeholder, and probably leads to too large fluxes over active biosphere, where COS mole fractions decline because of strong uptake. This is further discussed in Section 3.5."

The abstract was modified accordingly on Page 1 Lines 19-22 as:
"We finally find that the biosphere flux dependency on surface COS mole fraction (which was not accounted for in this study) may substantially lower the fluxes of the SiB4 biosphere model over strong uptake regions. Using COS mole fractions from our inversion, the prior biosphere flux reduces from 1053 Gg a-1 to 851 Gg a-1 which is closer to 738 Gg a-1 as was found by Berry et al. (2013)."

The discussion in Section 3.5 was also modified to reflect the information on Page 16 Lines 495-497 as:
"This simple correction, based on monthly mean fields, changes the biosphere sink from 1053 Gg a-1 to 851 Gg a-1 an update of 202 Gg a-1 (Supplementary Figures S12, S13 and S14), and closer to the 738 Gg a-1 reported by Berry et al. (2013)."

Minor Comments

Maybe adjust the abstract wording to slightly improve clarity that plant sink in the TM5 runs are zero order (not first order). For example, "We finally find that the biosphere flux dependency on surface COS mole fraction (which was not modeled in this study) may substantially. . ."

This has been fixed in the manuscript on Page 1 Line 20 as:
"We finally find that the biosphere flux dependency on surface COS mole fraction (which was not modeled in this study) may substantially lower the fluxes of the SiB4 biosphere model over strong uptake regions."

The introduction notes Suntharalingam et al. (2008) study which attempted to fit background data by increasing the plant sink but you may also want to reference the Campbell et al. (2008) finding that this upward revision could be validated using direct observations from the continental boundary layer from the intensive INTEX-NA airborne campaign. Campbell, J. Elliott, et al. "Photosynthetic control of atmospheric carbonyl sulfide during the growing season." Science 322.5904 (2008): 1085-1088.

We thank the reviewer to point out the publication. The citation of Campbell et al. (2008) was added on Page 2 Lines 53-54 as:
"Campbell et al. (2008) found that this upward revision could be validated using direct observations from the continental boundary layer from the intensive INTEX-NA airborne campaign."

The citation of Campbell et al. (2008) was also added in Section 3.5 Page 16 Line 488.

The authors site Lennartz et al. (2017) for the bottom-up ocean emissions but please also note the upward revision in Lennartz et al. (2019). Lennartz, S. T., von Hobe, M., Booge, D., Bittig,

H. C., Fischer, T., Gonçalves-Araujo, R., Ksionzek, K. B., Koch, B. P., Bracher, A., Röttgers, R., Quack, B., and Marandino, C. A.: The influence of dissolved organic matter on the marine production of carbonyl sulfide (OCS) and carbon disulfide (CS2) in the Peruvian upwelling, Ocean Sci., 15, 1071–1090, https://doi.org/10.5194/os-15-1071-2019, 2019.

We thank the reviewer to point out the publication. Citation to Lennartz et al. (2019) was added on Page 3 Line 58.

Section 2.1.1 should note the lack of observatories in the tropical continental boundary layer.

We agree. Section 2.1.1 Page 3 Lines 77-78 was modified as:
"It is worth to note that there is a lack of observations in the tropical continental boundary layer."

Regarding section 2.2.1, how did you divide the anthropogenic emissions into direct COS and indirect CS2? Did the Zumkehr data file present emissions separately for direct and indirect? If they didn't then how did you back this out? You may want to look at the emission inventory in Campbell et al (2015) which does present separate emission estimates for direct COS and indirect CS2. The Zumkehr approach was an extension of Campbell et al 2015.
Campbell, JE, Whelan, ME, Seibt, U, Smith, SJ, Berry, JA, and Hilton, TW (2015), Atmospheric carbonyl sulfide sources from anthropogenic activity: Implications for carbon cycle constraints. Geophys. Res. Lett., 42, 3004– 3010. doi: 10.1002/2015GL063445.

We implemented anthropogenic emissions largely based on Zumkehr et al. (2018) and Campbell et al. (2015). We managed to separate the COS emission from the Zumkehr et al. (2018) work according to Table 1 in Lee and Brimblecombe (2016). In Table 1 the authors reported a detailed emission budget for COS and $CS_2$ in various anthropogenic categories. Then we used the ratio of this budget to roughly estimate the direct and indirect COS anthropogenic emissions. In this way we were able to separate the COS and $CS_2$ direct anthropogenic emissions, which should be correct within the uncertainties. In Section 2.2.1 Page 6 Lines 150-152 were modified as:
"We applied a detailed anthropogenic emission budget for COS and $CS_2$ from Table 1 in (Lee and Brimblecombe, 2016). This allows us to roughly estimate the ratio of this budget and hence the direct and indirect COS anthropogenic emissions. The converted emissions averaged over the period 2000-2012 are summarized in Table 1."

We have also included the reference of (Campbell et al., 2015) in the manuscript, Page 5 Line 147.

Section 2.2.1 discussed uncertainties in molar yield. It might be worth noting that the uncertainty in the anthropogenic inventory is much larger than the uncertainty in molar yield.

We have clarified this in the revised manuscript. With more measurements available, we could try to reduce these uncertainties using our inverse modelling framework. The modification was made on Page 6 Lines 163-164 as:
"It is also worth noting that the uncertainty in the anthropogenic inventory is much larger than the uncertainty in molar yield."

Section 2.2.2 might want to draw comparison of this studies results to previous estimates from biomass burning in Campbell et al (2015) and open burning in Stinecipher et al. (2019).

We have discussed the differences in the revised manuscript. The major difference is that we considered new biofuel emission factors in South Asia. The modification was made on Page 6 Lines 172-176 as:

"Our biomass burning emissions in the 2000-2012 period are in the range 118-154 Gg a$^{-1}$ (Figure 2), similar to the emissions used in Berry et al. (2013) (135 Gg a$^{-1}$ and estimates reported in Campbell et al. (2015) (116±52 Gg a$^{-1}$). The more recent biomass burning estimate from Stinecipher et al, (2019) based on GFED 1997-2016 data reports global emissions of 60±37 Gg a$^{-}$1. Note, however, that biofuel use is not included in this estimate."

Regarding the poor posterior fit at NWR and THD, are there references in the CO2 inversion literature that had the same difficulty? These sites are designed to capture background mixing ratio but sometimes they suffer from local influence which would be one reason for the poor posterior fit. One helpful paper you may want to reference is Riley et al.
Riley, W. J., Randerson, J. T., Foster, P. N., and Lueker, T. J. (2005), Influence of terrestrial ecosystems and topography on coastal CO2 measurements: A case study at Trinidad Head, California, J. Geophys. Res., 110, G01005, doi:10.1029/2004JG000007.

We thank the reviewer to point out the $CO_2$ paper concerning THD. It is true that THD as discussed in the paper of Riley et al. (2005) is more affected by the local coastal effect and the diurnal cycle of $CO_2$ fluxes. The current TM5 inverse modelling effort applies a coarse resolution of $6° \times 4°$ globally, and thus the effect of costal meteorology at THD is not well captured. Another point is that biosphere flux of COS is applied on a monthly-average basis, and does not account for a diurnal cycle. The NWR station is probably also affected by local land effects that are not well resolved in the coarse simulations. To compare with $CO_2$ inversions, we have investigated CarbonTracker North America data and found similarities between COS and $CO_2$ inversions. For example, at THD, $CO_2$ has relatively large model-data mismatches of ~11 ppm in Summer (Figure A4). For NWR, the $CO_2$ model-data mispatch is about 3.37 ppm in Summer (Figure A5). In comparison, at MHD the $CO_2$ model-data mismatch is only 2.04 ppm in Summer inferred from Flask observations. Note that CarbonTracker North America employs a resolution of $1° \times 1°$ degree, compared to $6° \times 4°$ degree in our COS study.

Section 3.1 Page 11 Lines 326-327 was modified as:
"The local coastal effect might be another reason why THD yields a larger $\chi^2$ (Riley et al. 2005)."

[Figure]

*Figure A4. CO₂ observation and simulation by CarbonTracker at THD (source: https://www.esrl.noaa.gov/gmd/ccgg/carbontracker/co2tser.php?ds=co2_thd_surface-flask_1_representative&ed=assim&lastds=co2_nwr_surface-flask_1_representative).*

[Figure]

*Figure A5. CO₂ observation and simulation by CarbonTracker at NWR (source: https://www.esrl.noaa.gov/gmd/ccgg/carbontracker/co2tser.php?ds=co2_nwr_surface-pfp_1_allvalid-3magl&ed=assim&lastds=co2_nwr_surface-insitu_3_nonlocal ).*

Line 421: "However, observations clearly show a large drawdown of COS near the surface (Hilton et al., 2017; Spielmann et al., 2020)." You may want to reference the INTEX-NA data which is an intensive sampling of vertical profiles in the continental boundary layer (Campbell et al.,2008)

We have included the reference for (Campbell et al., 2008) on Page 16 Line 488.
* * *
Comments

The uptake of OCS is tied to the OCS concentration within the canopy. There are large variations in OCS uptake as OCS depleted air flows through vegetation, e.g. Berkelhammer et al. 2020 (https://doi.org/10.1029/2019GL085652). Most canopies do not see free troposphere concentrations of OCS. This will affect the plant uptake flux component substantially and should be addressed.

This comment was also made by reviewer #1. Because we use an inversion framework, we employ a zero-order flux approach, which is technically easier to implement. Indeed, we find adjustments that point to a large drawdown of COS in the canopy. Like we mentioned in the discussion, in analyzing the flux adjustment, we should be aware that part of the adjustment is due to the "concentration" effect. We have also cited Berkelhammer et al. (2020) on Page 16 Line 488.

DMS should not be considered a major source of OCS. Many researchers still refer to the 7% yield figure from the Barnes 1996 paper, but note that chamber studies proceeded without NOx and at high DMS concentrations. Subsequent studies demonstrated that an alternative chemical pathway is typically taken, and that changes in NOx affect OCS formation greatly (e.g. https://doi.org/10.1016/S1352-2310(98)00120-4). In other words, if you would like to include DMS, it is important to also model NOx. This is most likely such a small contribution that in the Whelan et al., 2018 synthesis, it was concluded that DMS should only be included as a source of uncertainty in the ocean flux rather than a source itself.

Currently we do not simulate NOx chemistry in the COS inverse modelling, since this largely increases the complexity of the modelling system. However, DMS is normally emitted over remote oceans, where low NOx concentrations prevail. We do not fully agree with the reviewer that the COS yield of 0.7% should not be included as long as the COS budget is not closed. The DMS as COS source is estimated as 162 Gg a$^{-1}$, and it is substantial compared with COS total budgets. Nevertheless, we have performed a NO-DMS inversion in which we placed the DMS-related COS emissions to the "unknown" category, with additional amount of 162 Gg a$^{-1}$. It can be seen that NO-DMS inversion shows similar pattern with a standard inversion Su (Figure A6 a-c). The difference is that DMS has been added to the optimized "missing" sources as shown in Figure A6 d.

We have modified Section 3.5 Page 16 Lines 472-475 as:
"Regarding DMS as COS precursor, we have evaluated its importance by performing a NO-DMS inversion, in which DMS as a tracer was removed and the 162 Gg a$^{-1}$ DMS source was added to the COS "unknown flux" in inversion Su. In Supplement Figure S11, it can be seen that the NO-DMS inversion shows larger adjustment over both oceans and continents, but that the pattern remains comparable to inversion *Su*."

[Figure]

*Figure A6. An analysis of No-DMS inversion compared with standard inversion scenario Su. (a) COS unknown posterior flux from inversion without DMS. (b) COS unknown posterior flux from inversion Su. (c) The difference between (a) and (b). (d) The annual mean of DMS source as COS in oceans.*

If the lifetime of CS2 was 12 days, it might make sense to model CS2 separately, since the associated OCS will not show up in the air parcel until it has traveled nearly around the globe. However, more recent evidence suggested that the lifetime is much shorter than that. For example, see the 3D atmospheric transport study performed by Anwar Khan which focusses only on CS2 and estimates a lifetime of less than 4 days at maximum: 10.3934/environsci.2017.3.484.

We thank the reviewer for pointing out this more recent paper regarding the $CS_2$ lifetime. We implemented the $CS_2 + OH$ chemistry using the rate quoted in (Khan et al. 2017). As a result, we spotted a mistake in the rate constant of the $CS_2 + OH$ reaction in the Khan et al. (2017) paper. According to (Khan et al. 2017), the rate constant of $CS_2+OH$ is (their Table 1):

$$rate\ constant\ 1 = \frac{1.25 \times 10^{-16} e^{\frac{4550}{T}}}{(1 + 1.81 \times 10^{-13} e^{\frac{3400}{T}})}$$

However, the rate constant (Sander et al., 2006, Hynes et al., 1988) should read:

$$rate\ constant\ 2 = \frac{1.25 \times 10^{-16} e^{\frac{4550}{T}}}{(T + 1.81 \times 10^{-3} e^{\frac{3400}{T}})}$$

We verified with the authors of Khan et al. (2017) that this is a typo and that the rate constant is correctly implemented in their model. We also implemented the Sander et al. (2006) rate constant in our simulations. Results are shown in Figure A7.

[Figure]

*Figure A7: Monthly averaged burden (left) and atmospheric lifetime (right) of CS2, calculated in TM5 with the rate constant of Sander et al. (2006) implemented.*

As can be seen in Figure A7, the burden and lifetime vary considerably over the year. The yearly average lifetime (burden/loss) amounts to 6.2 days, substantially larger than the value quoted in Khan et al. (2017): 2.8-3.4 days. Possible reasons are (1) our burden is higher due to larger emissions and possibly lower OH (2) no deposition has been implemented in our simulations. In Khan et al. (2017) loss through deposition in their standard run account for ~15% of the $CS_2$ removal. Note also that their standard model simulation underestimated $CS_2$ mole fractions, specific over large emission regions.

For comparison, Figure A8 shows the results from our standard run, with a constant $CS_2$ + OH rate from Jones et al. (1982): $2.0 \times 10^{-12}$ $cm^3$ molecules$^{-1}$ s$^{-1}$.

[Figure]

*Figure A8: Monthly averaged burden (left) and atmospheric lifetime (right) of CS2, calculated in TM5 with the rate constant of Jones et al. (1982) implemented.*

As can be seen, the lifetime and burden are slightly larger using this rate. The yearly average lifetime amounts to 9.4 days. Figures A9 and A10 show the January and July COS mole fraction difference at the surface between our standard model and the Sander et al. (2006) rate. It can be seen that the differences are relatively small and remain smaller than 10 pmol mol$^{-1}$.

Since the lifetime in TM5 is relatively long (> 10 days in the Northern Hemisphere winter), we would argue that it is necessary to simulate $CS_2$ as a separate tracer, to model the delayed COS production from $CS_2$ oxidation. To highlight this further, we have (in the Supplement of the revised paper) shown results from a simulation in which we emit $CS_2$ directly as COS. The modification was made on Page 15-16 Lines 463-472 as:

"We have also considered some variations in our modelling setup. A unique approach of our study is the inclusion of $CS_2$ and DMS as COS precursors. We tested the effect of emitting $CS_2$ ocean and anthropogenic sources directly as COS in an additional forward model simulation. As shown in Figure S10, COS mole fractions would become significantly larger close to $CS_2$ emission hot spots in Asia, Europe and the US. At selected stations (LEF in the US and MHD

in Europe, Figure S10 a and b), we observe COS mole fractions that are up to 40 pmol mol[-1] higher during events where emitted $CS_2$ is advected to the station. Some ambiguity has been introduced about the $CS_2$ lifetime (Khan et al., 2017). In our Su inversion, the lifetime of $CS_2$ is estimated as 9.4 days ($CS_2$ burden divided by $CS_2$ loss by OH), substantially longer than ~3 days lifetime mentioned in Khan et al. (2017). Future work should be based on the rate recommendations in Sander et al. (2006). Thus, we conclude that inclusion of $CS_2$ as a separate tracer is important if we want to understand emissions of $CS_2$ and COS which have distinctly different spatial patters (e.g. see Supplementary Figure S4.)"

[Figure]

Figure A9. COS mole fractions difference between this study and the rate as (Sander et al., 2006). This is the average COS mole fraction difference in January 2008-2010. The maximum and minimum values of the difference are marked in the left bottom corner.

[Figure]

Figure A10. The same as Figure A9 but for July 2008-2010.

The SiB3 model had a known phenology problem, where the growing season starts too soon (by a couple of weeks, perhaps) and ends too early. Does SiB4 have this issue? It would be good to check the seasonal timing by comparing to SiB3 or even another slightly complicating proxy, e.g. SIF.

Our biosphere fluxes are based on simulations with the Simple Biosphere model, version 4 (SiB4) (Haynes et al., 2019). New in SiB4 compared to SiB3 (used by Berry et al. (2013)) are

capabilities to simulate i.e. carbon pools, land cover heterogeneity, and leaf phenology. Further, COS uptake formulations in SiB4 are the same as in Berry et al. (2013). One of the new features in SiB4 compared to SiB3 is that it includes prognostic phenology, and this phenology is no longer depending on satellite products. So, if SiB3 had a problem with the phenology this does not automatically mean that SiB4 has the same problem. In fact, from an ongoing comparison with field observations the phenology looks good in SiB4. Results will be presented in a manuscript that is currently in preparation by co-author Linda Kooijmans. Therefore, we consider further discussion of the prior SiB4 fluxes beyond the scope of this study.

The abstract should be revisited to better reflect the conclusions of the study and to tidy up the language, e.g. sources of OCS are obviously included in current budgets.

We agree. The abstract has been revised to better show the conclusions and the cohesion of language. The modifications have been made above.

In short, revising the inclusion of DMS and addressing the first order plant uptake issue will certainly create a different budget overall, and may affect the conclusions.

We have discussed these issues in the revised manuscript in above comments. We argue, however, that a zero-order approach is sufficient for a first inverse modelling study, since the fluxes are allowed to adjust to the concentration effect as discussed in Section 3.5. Concerning DMS, we have also tested a NO-DMS inversion, which turns out to be consistent with a standard inversion scenario *Su*.

***References***:

Berkelhammer, M., Alsip, B., Matamala, R., Cook, D., Whelan, M. E., Joo, E., ... & Meyers, T. (2020). Seasonal Evolution of Canopy Stomatal Conductance for a Prairie and Maize Field in the Midwestern United States from Continuous Carbonyl Sulfide Fluxes. Geophysical Research Letters, 47(6), e2019GL085652.

Berry, J., Wolf, A., Campbell, J. E., Baker, I., Blake, N., Blake, D., ... & Stimler, K. (2013). A coupled model of the global cycles of carbonyl sulfide and CO2: A possible new window on the carbon cycle. Journal of Geophysical Research: Biogeosciences, 118(2), 842-852.

Haynes, K. D., Baker, I. T., Denning, A. S., Stöckli, R., Schaefer, K., Lokupitiya, E. Y., & Haynes, J. M. (2019). Representing grasslands using dynamic prognostic phenology based on biological growth stages: 1. Implementation in the Simple Biosphere Model (SiB4). Journal of Advances in Modeling Earth Systems, 11(12), 4423-4439.

Hynes, Anthony J., P. H. Wine, and J. M. Nicovich. "Kinetics and mechanism of the reaction of hydroxyl with carbon disulfide under atmospheric conditions." The Journal of Physical Chemistry 92.13 (1988): 3846-3852.

Jones, B. M. R., et al. "OCS formation in the reaction of OH with CS2." Chemical Physics Letters 88.4 (1982): 372-376.

Khan, A., Razis, B., Gillespie, S., Percival, C., & Shallcross, D. (2017). Global analysis of carbon disulfide (CS2) using the 3-D chemistry transport model STOCHEM. Aims Environ. Sci, 4, 484-501.

Khalil, M. A. K., & Rasmussen, R. A. (1984). Global sources, lifetimes and mass balances of carbonyl sulfide (OCS) and carbon disulfide (CS2) in the earth's atmosphere. Atmospheric Environment (1967), 18(9), 1805-1813.

Lee, C. L., & Brimblecombe, P. (2016). Anthropogenic contributions to global carbonyl sulfide, carbon disulfide and organosulfides fluxes. Earth-science reviews, 160, 1-18.

Sander SP, Friedl RR, Golden DM, et al. (2006) Chemical kinetics and photochemical data for use in atmospheric studies. Evaluation number 15, JPL publication 06-2, Jet Propulsion Laboratory, Pasadena, CA.

Zumkehr, A., Hilton, T. W., Whelan, M., Smith, S., Kuai, L., Worden, J., & Campbell, J. E. (2018). Global gridded anthropogenic emissions inventory of carbonyl sulfide. Atmospheric Environment, 183, 11-19.